# The CAMELS-CL dataset: catchment attributes and meteorology for large sample studies – Chile dataset

Camila Alvarez-Garreton[1,2], Pablo A. Mendoza[3,4], Juan Pablo Boisier[1,5], Nans Addor[6], Mauricio Galleguillos[1,7], Mauricio Zambrano-Bigiarini[1,8], Antonio Lara[1,2], Cristóbal Puelma[1,7], Gonzalo Cortes[9], Rene Garreaud[1,5], James McPhee[3,4], Alvaro Ayala[10,11]

[1]Center for Climate and Resilience Research (CR2), Santiago, Chile
[2]Instituto de Conservación, Biodiversidad y Territorio, Universidad Austral de Chile, Valdivia, Chile
[3]Department of Civil Engineering, Universidad de Chile, Santiago, Chile
[4]Advanced Mining Technology Center, Universidad de Chile, Santiago, Chile
[5]Department of Geophysics, Universidad de Chile, Santiago, Chile
[6]Climatic Research Unit, School of Environmental Sciences, University of East Anglia, Norwich, UK
[7]Faculty of Agronomic Sciences, Universidad de Chile, Santiago, Chile
[8]Department of Civil Engineering, Faculty of Engineering and Sciences, Universidad de La Frontera, Temuco, Chile
[9]Department of Civil and Environmental Engineering, University of California, Los Angeles, California, USA
[10]Laboratory of Hydraulics, Hydrology and Glaciology (VAW), ETH Zurich, Zurich, Switzerland
[11]Swiss Federal Institute for Forest, Snow and Landscape Research (WSL), Birmensdorf, Switzerland

*Correspondence to*: Camila Alvarez-Garreton (camila.alvarez@uach.cl)

**Abstract.** We introduce the first catchment dataset for large sample studies in Chile. This dataset includes 516 catchments, it covers particularly wide latitude (17.8 to 55.0 °S) and elevation (0 to 6993 m a.s.l.) ranges, and it relies on multiple data sources (including ground data, remote-sensed products and reanalyses) to characterise the hydroclimatic conditions and landscape of a region where in situ measurements are scarce. For each catchment, the dataset provides boundaries, daily streamflow records and basin-averaged daily time series of precipitation (from one national and three global datasets), maximum, minimum and mean temperatures, potential evapotranspiration (PET; from two datasets), and snow water equivalent. We calculated hydro-climatological indices using these time series, and leveraged diverse data sources to extract topographic, geological and land cover features. Relying on publicly available reservoirs and water rights data for the country, we estimated the degree of anthropic intervention within the catchments. To facilitate the use of this dataset and promote common standards in large-sample studies, we computed most catchment attributes introduced by Addor et al. (2017) in their Catchment Attributes and MEteorology for Large-sample Studies (CAMELS) dataset, and added several others.

We used the dataset presented here (named CAMELS-CL) to characterise regional variations in hydroclimatic conditions over Chile, and to explore how basin behaviour is influenced by catchment attributes and water extractions. Further, CAMELS-CL enabled us to analyse biases and uncertainties in basin-wide precipitation and PET. The characterisation of catchment water balances revealed large discrepancies between precipitation products in arid regions, and a systematic precipitation underestimation in headwater mountain catchments (high elevations and steep slopes) over humid regions. We evaluated PET products based on ground data and found a fairly good performance of both products in humid regions ($r > 0.91$) and lower correlation ($r < 0.76$) in hyper-arid regions. Further, the satellite-based PET showed a consistent overestimation of observation-based PET. Finally, we explored local anomalies in catchment response by analysing the relationship between hydrological

signatures and an attribute characterising the level of anthropic interventions. We showed that larger anthropic interventions are correlated with lower than normal annual flows, runoff ratios, elasticity of runoff with respect to precipitation, and flashiness of runoff, especially in arid catchments.

CAMELS-CL provides unprecedented information on catchments in a region largely underrepresented in large-sample studies. This effort is part of an international initiative to create a multi-national large sample datasets freely available for the community. CAMELS-CL can be visualised from http://camels.cr2.cl and downloaded from https://doi.pangaea.de/10.1594/PANGAEA.894885.

## 1 Introduction

Large-sample hydrology has been recognised as a fundamental framework to advance hydrological science (e.g., Andréassian et al., 2006; Ehret et al., 2014). The insights provided by studying a large set of catchments complement the findings from intensive place-based studies, where more detailed analyses are conducted over a small number of catchments. A common approach in large-sample studies is to explore interrelationships between catchment attributes describing landscape, climate and hydrologic behaviour, typically obtained from topography, soil types, geology, land cover and hydro-meteorological datasets (e.g., Oudin et al., 2008; Sawicz et al., 2011; Gupta et al., 2014; Newman et al., 2015; Addor et al., 2017). Accounting for catchments attributes in a comprehensive dataset serves various purposes. For example, comparative hydrology and catchment classification studies use these attributes to explore catchment (dis)similarities (e.g., McDonnell and Woods, 2004; Wagener et al., 2007; Sawicz et al., 2011; Berghuijs et al., 2014). Likewise, regionalisation studies incorporate catchment attributes to identify (hydro-climatically and physically) similar catchments that can be used to transfer model information from gauged to ungauged locations (Blöschl et al., 2013; Sawicz et al., 2011) – a fundamental motivation of the Predictions in Ungauged Basins (PUB) initiative (Sivapalan et al., 2003). In summary, the main goal of large-sample applications is to learn from diversity in order to define generalizable rules that can help to improve the predictability of the water cycle. This is addressed by disentangling the interplay between landscape, climate and hydrologic behaviour, which provides insights on hydrological systems and on suitable model structures to represent them.

As highlighted by Gupta et al. (2014), a key challenge in large-sample hydrology is data accessibility, which is particularly critical in data-scarce regions such as South America (see Fig. 2 in Gupta et al., 2014). Although there is a tendency for large-sample datasets to be shared worldwide (see examples in Gupta et al., 2014), available hydro-meteorological records from different countries typically use different formats and come from different providers. Moreover, they are rarely spatially aggregated to the catchment scale, which makes it difficult for researchers and practitioners to use them for basin-oriented applications.

In this paper, we introduce a unique dataset that includes 516 catchments in Chile, and show how this dataset serves to improve our understanding of hydrological systems and their predictability through the assessment of (1) the uncertainties in two key meteorological variables (precipitation and PET), and (2) the impacts of anthropic intervention on catchment response.

The dataset built here consists on catchment boundaries in shapefile format, hydro-meteorological time series, and a suite of catchment attributes based on climate, hydrology, topography, geology, land cover, and water use. To facilitate and encourage the use of this dataset, and to promote common standards and formats in large-sample studies, we compute five (out of six) classes of catchment attributes (location and topography, geology, land cover characteristics, climatic indices and hydrological signatures) used in Addor et al. (2017, referred as A17 hereafter). A17 introduced the Catchment Attributes and MEteorology for Large-sample Studies dataset (CAMELS dataset), which encompasses meteorological and streamflow datasets collated by Newman et al. (2015) and provides quantitative estimates of a wide range of attributes for 671 catchments in the contiguous United States (CONUS). The CAMELS dataset has already been used in a myriad of applications, including assessment of streamflow skill elasticity to initial conditions and climate prediction (Wood et al., 2016), snow data assimilation for seasonal streamflow prediction (Huang et al., 2017), continental-scale hydrologic parameter estimation (Mizukami et al., 2017), and climate change impacts on the hydrology of the CONUS (Melsen et al., 2018), among others. Following this nomenclature, we name our dataset CAMELS-CL, which stands for CAMELS dataset in Chile. We add an attribute class not covered by A17: the degree of human intervention in each catchment. This novel information is valuable since anthropogenic activities may have major impacts on catchment behaviour, but human influence is often difficult to quantify, especially for hundreds of catchments.

We characterise hydrological systems in Chile by analysing the spatial distribution of catchments attributes provided in CAMELS-CL. Subsequently, we apply CAMELS-CL to assess uncertainties in precipitation and potential evapotranspiration estimates, and to quantify anthropic impacts on catchment response. To this end, we compare the different precipitation products and evaluated them based on the observed water balance. This analysis includes one national dataset (CR2MET) and three widely used global datasets (CHIRPS, MSWEP and TMPA); thus, the results may have implications beyond the domain covered by CAMELS-CL. Then, we assess PET products based on an independent set of PET point values calculated from meteorological records. Finally, we analyse human influence on catchment behaviour by relating hydrological signatures with a human intervention attribute calculated from water extractions information.

The paper is structured as follows: Sect. 2 describes the study area; Sect. 3 describes the collected datasets (Sect. 3.1) and provides a description of the derived catchment attributes with a discussion of their spatial distribution (Sect. 3.2); Sect. 4 presents the precipitation (Sect. 4.1) and potential evapotranspiration (Sect. 4.2) uncertainty analyses; Sect. 5 presents the analysis of human influence on catchment behaviour; and Sect. 6 summarises the main conclusions of the paper.

**2 Study area**

The area covered by CAMELS-CL corresponds to continental Chile, a territory with a distinct geographical configuration that spans 4300 km along a north-south axis over the west of South America (17.8° S to 55.0° S). The country lies on the Nazca and Antarctic tectonic plates. The tectonic activity in the Quaternary (early Pleistocene) led to the formation of the three main physiographic characteristics of the territory (from west to east): the coastal range, the intermediate depression, and the Andes

Cordillera. Featuring altitudes well above 3000 m a.s.l., with summits up to 7000 m a.s.l. (e.g. Aconcagua mountain or Ojos del Salado volcano), the Andes acts as an effective barrier for atmospheric flows, leading to particularly large precipitation amounts at high elevations (Garreaud, 2009) and to a noticeable contrast between the rainfall regimes of southern Chile (wet) and of the Argentinean Patagonia (dry).

Chile has 16 administrative regions (Fig. 1) split into four macro-zones defined by the Chilean Water Directorate (DGA), based on hydrological, climatic and topographic features (DGA, 2016a): North (from Arica and Parinacota to Coquimbo regions); Central (from Valparaiso to Maule regions); South (from Bio-Bio to Los Lagos regions); and Austral (from Aysén to Magallanes regions). To provide a more detailed description, we divided the North macro-zone into Far North (from Arica and Parinacota to Antofagasta regions) and Near North (from Atacama to Coquimbo regions), and the Austral macro-zone into

Austral zone (Aysen region) and Southern Patagonia (Magallanes region). The resulting six macro-zones are presented in Fig. 1.

The country includes five primary climatic regimes according to the Köppen-Geiger climate classification (Kottek et al., 2006; Sarricolea et al., 2017). The Far North is dominated by a cold desert climate (BWk) and tundra (ET) along the Andes range. The Near North is characterised by cold desert climate in the Atacama region and a cold semi-arid climate (BSk) in the

Coquimbo region. The Central zone is dominated by a sub-humid Mediterranean climate (Csb). The Southern zone includes a humid Mediterranean climate in the Bio-Bio and Araucanía regions, and a temperate rain-oceanic climate (Cfb) in Los Rios and Los Lagos regions. The Austral and Southern Patagonia zones are dominated by rain-cool oceanic (Cfc) and cold steppe (BSk) climates.

## 3 CAMELS-CL dataset

**3.1 Input data**

### 3.1.1 Topography and catchment boundaries

The first step in the development of CAMELS-CL was the delimitation of catchment boundaries (Fig. 2). An official database for Chilean hydrographic network was developed by the Instituto Geográfico Militar in 1984 (IGM, 1984) and updated by the DGA in 2014 (DGA; CIREN, 2014). This network delineation followed the Strahler hierarchy (Strahler, 1957), using the 30 m ASTER GDEM elevation data (Tachikawa et al., 2011). The DGA network includes 102 catchments, 491 sub-catchments

m ASTER GDEM elevation data (Tachikawa et al., 2011). The DGA network includes 102 catchments, 491 sub-catchments and 1481 sub-sub-catchments, and has been largely used by government agencies, the private sector and the general public. However, a key limitation of this hydrographic network is that – given the methodology used for its implementation – the existing streamflow gauges do not correspond with catchment outlets. Furthermore, DGA catchment boundaries are truncated at the administrative national border, even though some catchments contribute with runoff from Bolivian and Argentinian

territories. Since any hydrologic application within a controlled basin requires the total contributing area associated to

streamflow measurements, and there is no official catchment boundary database, different studies have applied their own basin delineations, making it difficult to compare results.

To overcome this limitation, we created our own catchment boundaries database for CAMELS-CL, defining the basin outlets at the location of 516 selected streamflow gauges (Sect. 3.1.5), and following only topographic-driven limits (not the

administrative national border). A key challenge for this task is the mismatch between some station geographic coordinates reported by the DGA and the actual river network location – detected through the inspection of Google Earth imagery (Google, 2016). For some of those cases, expert advice was obtained from DGA technicians regarding gauge locations, while, for others, ancillary information (e.g., gauge name, road maps, Google Earth imagery) was used to determine the most likely location. Basin delineation was performed in Quantum GIS (QGIS Development Team, 2015) by using watershed delineation packages

from the Geographic Resources Analysis Support System (GRASS) (Neteler et al., 2012) and 30 m ASTER GDEM as input elevation data. Given the location of streamflow gauges, several catchments collected in this dataset are nested. We report this by using a logical hierarchy matrix indicating which basins are contained within another catchment of the dataset. The hierarchy matrix can be used to filter independent catchments, which is required for some applications such as hydrological modelling of large basins, catchment classification and parameter regionalisation.

Main topographic properties including area, median, mean, minimum and maximum elevation, and mean slope were computed for each catchment from ASTER GDEM 30 m raster data, clipped by the catchment boundary polygons and processed with the R raster package (Hijmans, 2016). An important limitation of this dataset is that its spatial resolution is relatively coarse, which can lead to errors when delineating catchments over very flat regions (such as the Far North, see Sect. 3.2.1).

### 3.1.2 Geology

Catchment-scale geological characteristics were retrieved from the Global Lithological Map database (GLiM) produced by Hartmann and Moosdorf (2012). GLiM is a compilation of national datasets into a unified global map. In the case of Chile, GLiM uses the map produced by the Servicio Nacional de Geología y Minería (Sernageomin, 2004), which has a resolution of 1:1000000 and is the most complete and commonly used map for the country. For each catchment, we reported the most and second-most frequent geological class, as well as the fraction of the catchment they cover. We also extracted the fraction

of the catchment described as "carbonate sedimentary rocks", as it is a useful indicator of the presence of karstic systems.

### 3.1.3 Land cover

We used the 30 m resolution land cover map provided by Zhao et al. (2016), which integrates multi-seasonal Landsat 8 imagery acquired during 2013 and 2014, complemented with Moderate Resolution Imaging Spectro-radiometer (MODIS) Enhanced Vegetation Index data, and high resolution imagery on Google Earth. The classification scheme adopted by Zhao et al. (2016)

was designed with Chilean geographers and biodiversity researches, based on the FROM-GLC project (Gong et al., 2013), which is similar to the Land Cover Classification System (Di Gregorio and Jansen, 2005). This classification scheme is compatible with other land cover classification systems such as FAO or IGBP, with minor ancillary data. It consists of 10 main

(level-1) classes (Fig. 1): croplands; forests, grasslands; shrublands; wetlands; water bodies; impervious surfaces; barren lands; and snow and ice. Some classes were refined in level-2 (e.g., forests were separated in native forest and exotic forest plantation) and level-3 subclasses (a total of 30 and 35 subclasses, respectively). For CAMELS-CL, we used the R raster package (Hijmans, 2016) to clip the land cover map within each catchment boundary polygon, and compute the fractional area 5   associated with each class or subclass (as described in Table 3).

### 3.1.4 Glaciers

Glaciers in Chile can be found at several locations, varying from small ice bodies at high-elevation sites in the Atacama region, to alpine glaciers in the Central Zone, and the large Patagonian ice fields in the Austral and Southern Patagonia regions (Pellicciotti et al., 2014). Even though the land cover map from Zhao et al. (2016) identifies areas of snow and ice, we included 10   a global glacier inventory for calculating the degree of glacierisation of the selected catchments. Glacier inventories have the advantages of using geomorphologic glacier-delineation techniques, and the recognition of debris-covered areas that cannot be identified by land cover classification schemes. In this study, we used the latest version of the Randolph Glacier Inventory (RGI 6.0; RGI Consortium, 2017). RGI 6.0 is a globally complete inventory of glacier outlines and it is widely used in regional and global studies on land surface fluxes, climatology and meteorology (e.g., Huss and Hock, 2015; Marzeion et al., 2012; 15   Mernild et al., 2017). We preferred to use RGI 6.0 rather than the Chilean glacier inventory from DGA (DGA, 2014) because there are portions of some catchments lying on Argentinean territory (Fig. 2). The RGI 6.0 was clipped within each catchment and two attributes were computed: the total glacierised area ($km^2$) and the percentage of glacierised area in the catchment (%).

### 3.1.5 Streamflow

We compiled daily streamflow records for gauges maintained by the DGA, available from the CR2 Climate Explorer 20   (http://explorador.cr2.cl/). From the 809 gauges included there, we selected those currently operational (independently of their data period), or suspended after 31 December 1980 with a record period longer than 10 years. We also discarded gauges located in artificial channels, ending up with 516 selected gauges. The record lengths from the selected gauges range from 192 to 366667 days, with a mean (median) of 10979 (9909) days. Figure 3 illustrates the availability of daily streamflow records for different time periods (represented with different colours), which is mainly due to the different record periods of the selected 25   gauges. Further, sparse missing daily records may be found in streamflow time series due to specific extreme events, where the station might have not worked properly or might have been broken. Note that hydrological year is considered from April $1^{st}$ to March $31^{st}$. As expected, the number of stations decreases with longer data availability. For example, if only stations with at most 5 % of missing data were selected, this would lead to a subset of 90 to 115 stations (which corresponds to 18 % and 22 % of the total number of catchments within the database, respectively), depending on the time period selected. When 30   considering all stations with at most 30 % of missing data, then 249 to 258 stations (48 % and 50 % of catchments, respectively) would meet this criterion, depending on the period (Fig. 3). Figure 4a presents the mean annual discharge for each station (computed for the entire record period).

### 3.1.6 Precipitation

In most cases, precipitation is the main driver of hydrological systems. However, the geographical distribution of this variable is highly uncertain, even in densely monitored regions (Tian and Peters-Lidard, 2010; Woldemeskel et al., 2013). This limitation is aggravated in regions with difficult accessibility, where only a sparse network of meteorological stations is available. In order to account for robust precipitation estimates and to characterise the uncertainty of this variable, we processed catchment-scale precipitation from four different products, whose main characteristics are summarised in Table 1. Daily precipitation fields provided by each product were clipped and averaged within the catchment boundaries, resulting in four daily time series for each catchment, named $precip_{cr2met}$, $precip_{chirps}$, $precip_{mswep}$ and $precip_{tmpa}$.

The $precip_{cr2met}$ times series was derived from CR2MET, a spatially-distributed daily precipitation product developed for Chile, which is currently used by DGA to update the national water balance (DGA, 2017). The CR2MET product is partly based on a statistically downscaling of ERA-Interim reanalysis data (Balsamo et al., 2015). The method builds on multiple linear regression models used to transfer precipitation, moisture fluxes and other variables from ERA-Interim onto 0.05º resolution precipitation estimates. The statistical models, which also consider a number of topographic parameters, were calibrated using a large network of quality-controlled rain-gauge records. Depending on the distance of a given grid cell to neighbouring stations, the final product was obtained from merging downscaled precipitation and spatially interpolated in situ observations. Further information about formulation, quality control and product assessments can be found in DGA (2017).

The three satellite-based precipitation products used in CAMELS-CL were selected following the inter-comparison reported by Zambrano-Bigiarini et al. (2017) for the entire Chilean territory. The $precip_{chirps}$ time series was computed from the Climate Hazards Group InfraRed Precipitation with Station data version 2 (CHIRPS, Funk et al., 2015), a long term (1981 to near present), quasi global (50° N to 50° S) daily dataset available at a spatial resolution of 0.05°, designed to monitor agricultural drought and global environmental changes over land. CHIRPS uses the Tropical Rainfall Measuring Mission Multi-Satellite Precipitation Analysis version 7 (TRMM 3B42v7) in order to calibrate global Cold Cloud Duration rainfall estimates (Funk et al., 2015). CHIRPS also incorporates surface rain-gauge data in order to reduce estimation biases, based on public and private monthly data. Originally, this dataset spanned from 50º N to 50º S, but since November 2012 data is not being produced south of 46º S. More information can be found in Funk et al. (2015).

The variable $precip_{mswep}$ was computed from the Multi-Source Weighted-Ensemble Precipitation (MSWEP, Beck et al., 2017) data version 1.1, a fully global precipitation dataset released in June 2016, with a 3-hourly temporal and 0.25º spatial resolutions, specifically produced for hydrological modelling applications. MSWEP was designed to improve the performance of satellite products in representing precipitation over mountainous, tropical, and snowmelt-driven regions. The algorithm used in MSWEP merges observed rain-gauge data, satellite observations and reanalysis data to provide reliable precipitation estimates over the entire globe. In this paper, we used daily data from MSWEP version 1.1, but newer versions (already available) will be included in CAMELS-CL after validation with ground measurements in Chile.

Finally, precip$_{tmpa}$ was computed from the Tropical Rainfall Measuring Mission (Huffman et al., 2007) Multi-satellite Precipitation Analysis (TMPA) dataset, which provides quasi global (50° N to 50° S) precipitation estimates 0.25° spatial resolution. TMPA integrates infrared and passive microwave data from a wide variety of satellite-borne precipitation-related sensors. In this study, we used the TRMM research product 3B42v7, which makes use of Global Precipitation Climatology Project (GPCP; Adler et al., 2003) and Climate Assessment and Monitoring System (CAMS, Ropelewski et al., 1984) data to rescale its estimations on a monthly basis.

### 3.1.7 Temperature

Daily time series of minimum, maximum and mean temperatures for each catchment were also derived from the CR2MET dataset (DGA, 2017). Daily minimum and maximum temperatures in CR2MET (CR2MET/T$_{max}$ and CR2MET/T$_{min}$, respectively) were mapped for the period 1979-2016 using a slightly different approach than the one used for precipitation (Sect. 3.1.6). In this case, the method used land-surface temperature (LST) estimates from MODIS satellite retrievals, in addition to near surface temperature provided by ERA-Interim. Multivariate regression models for both CR2MET/T$_{max}$ and CR2MET/T$_{min}$ were developed using LST as part of the explanatory variables and local temperatures records in Chile as target data. Given the data gaps and relatively short period available for LST, the final product was derived for the whole period (1979-2016) by fitting the ERA-Interim data to the preliminary (incomplete) MODIS-based product. To get mean daily temperatures (CR2MET/T$_{mean}$), the long-term CR2MET/T$_{max}$ and CR2MET/T$_{min}$ were used to adjust the ERA-Interim 3-hourly near surface temperature. The adjusted 3-hourly data was then averaged to derive CR2MET/T$_{mean}$. Gridded daily mean, minimum and maximum temperatures from CR2MET (0.05° lat-lon resolution) were clipped to obtain basin-averaged daily time series for CAMELS-CL, named T$_{mean}$, T$_{min}$ and T$_{max}$, respectively.

### 3.1.8 Potential evapotranspiration

We processed catchment-scale PET from two different sources. The first PET product uses the formulae proposed by Hargreaves and Samani (1985), which is solely based on surface temperature data (see Hargreaves and Allen, 2003 for further details). We used CR2MET/T$_{max}$ and CR2MET/T$_{min}$ (described in Sect. 3.1.7) to generate a gridded PET estimate (PET$_{har}$). The second PET data included in CAMELS-CL (PET$_{mod}$) is that provided by the MODIS PET product (MOD16 collection 5; Mu et al., 2005), which is processed from different sources of information, including leaf area index and fractional photosynthetically active radiation, FPAR/LAI (MOD15A2; Myneni et al., 2002), land cover type 2 (MOD12Q1; Friedl et al., 2002), albedo (MCD43B2 and MCD43B3; Jin et al., 2003; Lucht et al., 2000), and daily meteorological reanalysis data from NASA's MERRA GMAO (GEOS-5). MOD16 is calculated following the Penman-Monteith approach (Howell and Evett, 2001), and the final product is available at an 8 day temporal resolution for the period 2000-2014, on a 1×1 km$^2$ grid. Such as for other gridded variables, the PET$_{har}$ and PET$_{mod}$ products were clipped and averaged within basin boundaries to generate daily (called pet$_{har}$) and 8 day (called pet$_{mod}$) catchment-scale time series, respectively.

### 3.1.9 Snow water equivalent

We processed daily snow water equivalent (SWE) data using the 180 m resolution SWE product generated by Cortés and Margulis (2017), which covers the Near North and Central Zone (25-37º S). Cortés and Margulis (2017) obtained SWE ensemble estimates from forward modelling "prior" values, which were conditioned trough the assimilation of historical fractional snow-covered area (fSCA) data from Landsat TM, ETM+ and OLI sensors. The "posterior" SWE and fSCA estimates were probabilistically conditioned on the observed depletion record from Landsat, the uncertainty in fSCA observations, and the forward model state uncertainty. The fSCA retrieval was obtained with a spectral un-mixing algorithm (Cortés et al., 2014). The forward models for prior ensemble generation were the SSiB3 land surface model (Yang et al., 1997) and a Snow Depletion Curve model (SDC; Liston, 2004). Detailed assessments of this reanalysis framework were performed for the Sierra Nevada using in situ sensor data (Margulis et al., 2016), and for the Andes (Cortés et al., 2016) using snow survey points, site-years of peak annual snow pillow and snow course SWE observations. Validation results showed unbiased posterior SWE estimates with a correlation coefficient of 0.73, RMSE of 0.29 m and mean error less than 0.01 m using snow pillow and snow course peak SWE. Results using snow survey data showed similar unbiased estimates, with a correlation coefficient of 0.50, RMSE of 0.29 m and mean error less than 0.01 m. The daily SWE gridded product generated by Cortés and Margulis (2017) was clipped and averaged within the catchment boundaries to obtain daily time series for each catchment.

### 3.1.10 Water rights and reservoirs information

A public reservoir dataset (http://www.ide.cl/descarga/capas/item/embalses-2016.html, accessed on September 2017) was processed to identify the presence of dams within catchments. We also compiled and processed granted water rights available from the National Water Atlas (DGA, 2016a). This water allocation dataset includes information about the source (surface or groundwater), the type of right (i.e., consumptive or non-consumptive), use (i.e., industrial, irrigation, domestic and drinking water, hydroelectric power, pisciculture, mining, and classified as "other uses"), annual allocated flow (expressed in units of volume per time or as "shares"), and temporal allocation (i.e., permanent and continuous, permanent and discontinuous, permanent and alternated, eventual and continuous, eventual and discontinuous, or eventual and alternated). A detailed explanation of this water right classification can be found in Carey (2014). A key limitation of this dataset is the lack of information on the actual use of granted rights (Larraín, 2006). Additionally, some water right records have incomplete information (e.g., missing coordinates, water volume assigned and temporal allocation).

Figure 5 illustrates water allocation in central-southern Chile (30-43º S), showing surface and groundwater rights (all types). It is clear that groundwater rights dominate in the central Chile (31-36° S), especially in low elevation areas, compared to surface water rights. On the other hand, more surface water rights are granted in southern Chile, especially within high elevation areas towards the Andes.

**3.2 Derived catchment attributes**

We computed 70 catchment attributes grouped in six classes (Table 2). To motivate the use of common standards in the development of large sample catchment datasets, we included most of the attributes presented by A17. A comparative summary between CAMELS and CAMELS-CL attributes is presented in Table 2, from which one can note that the attributes from
classes climatic indices and hydrological signatures were fully adopted from A17. The attributes from the class soils characteristics were not computed at this stage since there is no publicly available national dataset. Given the differences in input datasets, some of the attributes from the classes location and topography, geologic characteristics, and land cover characteristics in A17, were not computed here. On the other hand, new attributes were derived for the classes location and topography (Sect. 3.2.1), land cover characteristics (Sect. 3.2.3) and hydrological signatures (Sect. 3.2.5). Further, a new class
was added to describe the degree of intervention within the catchments (Sect. 3.2.6).

A complete list of catchment attributes included in CAMELS-CL, their description and the corresponding data sources are presented in Table 3. To ensure the reproducibility of our results, the reference to the explicit formulation of climatic indices and hydrological signatures is provided in Table 3. Discussions on the spatial distribution of catchment attributes, separated by class, are presented in the following sub-sections.

**3.2.1 Location and topography**

Figure 6 shows six (out of 14) location and topography attributes. Figure 6a presents the elevation of catchment outlets, illustrating two main elevation gradients: (i) a north to south (N-S) decrease, starting with high elevation basins in the Far North macro-zone – which corresponds to the southern portion of the Altiplano plateau (18-22° S) (Allmendinger et al., 1997) –, towards lower elevations in the southern macro-zones; and (ii) an east to west (E-W) gradient, dominated by high elevations
in the Andes (located along the east border) decreasing towards sea level at the west border. This gauge elevation attribute can be used to classify catchments based on their location with respect to the coast or the Andes. We proposed the attribute location_type (see Table 3 and Fig. 6f) with three categories: coastal (or low elevation), foothills and Altiplano catchments, defined by gauge elevations lower than 50 m a.s.l., between 1000-1200 m a.s.l., and above 3500 m a.s.l., respectively.

Figure 6b reveals smoother N-S and E-W gradients of basin-averaged elevations, compared to gauge elevation gradients (Fig.
6a). This is because the mean elevation calculated for downstream catchments includes nested catchments (located at higher altitudes). From the complete set of catchments, 178 have a mean elevation greater than 2000 m a.s.l. The spatial distribution of mean catchment slopes follows different patterns depending on the macro-zone (Fig. 6c). The Far North – dominated by the flat Altiplano Plateau – exhibits relatively small variations in mean slopes, with relatively low values. From Near North to Austral Zone, the mean slope shows a spatial distribution similar to that from mean elevation, with a E-W gradient dominated
by high slopes in the Andes and flatter areas towards the sea. In southern Patagonia, such E-W gradient is reversed given the relative position of the Andes.

The spatial distribution of basin areas shows a general increase from east to west (Fig. 6d), which is consistent with smaller headwater catchments at the Andes, and larger downstream catchments towards the sea. Some exceptions to this E-W distribution pattern are catchments located near the east border, featuring either a N-S drainage direction, or a portion of their total contributing area in Argentina (beyond the east national border). Additional exceptions to such E-W distribution are small

inner sub-catchments near the west border, or small headwater catchments originated at the Chilean Coastal Range, which runs from north to south along the Pacific coast and reaches up to 3000 m a.s.l. in the Antofagasta region (Figueroa and Moffat, 2000).

Given that all catchments were delineated using available streamflow gauge locations as outlets (Sect. 3.1.1), the contributing area (Fig. 6d) is not necessarily correlated with the number of nested basins within each catchment (Fig. 6e). For example,

some small catchments might be highly instrumented (i.e., with many controlled nested basins, because of – for example – water allocation priorities or having high population density like those in Region Metropolitana, which concentrates more than 40% of the country population), while large but poorly instrumented catchments might not have inner basins defined.

### 3.2.2 Geological attributes

Overall, the most common dominant geological classes within CAMELS-CL catchments are acid plutonic rocks (24 %), acid

volcanic rocks (20 %) and pyroclastic (14 %). In the Far North zone, there is a strong presence of Pyroclastics, Siliclastic sedimentary rocks and Intermediate volcanic rocks (Fig. 7a and 7b), which can result in the connection of groundwater systems through fractured volcanic rocks (DGA, 1986). This means that there might be differences between surface catchment boundaries (based on topography) and the extension of groundwater systems, which should be considered when analysing basin-scale hydrological response. Figure 7a also indicates that strong geological differences may exist between neighbouring

catchments. Furthermore, one can see generally high geological variability within the catchments. Indeed, the dominant geological class covers less than half of the contributing area in most catchments, as indicated by the histogram in Fig. 7c. The presence of carbonate sedimentary rocks is particularly low (Figure 7e), with only 24 catchments having at least 10 % of this type of rock. This suggests low formation of karst, a subsurface characteristic featuring large fissures and voids, which results in fast infiltration rates and preferential permeability channels (La Moreaux et al., 1984).

### 25   3.2.3 Land cover attributes

As summarised in Table 2, five land cover attributes in A17 were not computed since the land cover map used here (from Zhao et al., 2016) does not provide information about leaf area index, green vegetation fraction, or depth. Instead, we included land cover attributes based on the catchment area encompassed by the main classes of the land cover dataset (Table 3). The first nine land cover attributes described in Table 3 were computed as the percentage of the catchment area covered by levels 1 and

2 land cover classes defined by Zhao et al. (2016). We also computed a forest plantation index to quantify the ratio between forest exotic plantation (mainly *Pinus radiata* and *Eucalyptus spp*) and native forest within a catchment, which is critical information for forest hydrology and ecosystems studies (e.g., Lara et al., 2009a).

Considering that several catchments (almost 50; Fig. 1) extend beyond the Chilean territory, a key limitation of the land cover attributes derived from Zhao et al. (2016), is the lack of information outside the national boundary. To address this, we generated an attribute indicating the fractional catchment area contained within the land cover map, serving also as a quality flag for basin-averaged land cover characteristics.

Another limitation of the land cover map and derived attributes is that there is no characterisation of inter-annual variability (the map was constructed by using imagery from 2013 and 2014). This can be particularly important for land cover types that are sensitive to climatic variations, such as altiplanic wetlands, which largely influence the hydrology in the Far North. This limitation is also critical for classes featuring drastic changes within time, such as forest plantation and cropland.

Figure 8 illustrates a subset of the land cover attributes listed in Table 3. Figure 8a shows the forested (native forests and forest
plantation types) catchment area, which prevails in the Southern Zone, Austral Zone and Southern Patagonia. In forested catchments, exotic forest plantations dominate the coastal areas of the Central and Southern Zones, with forest plantation indices up to one (Fig. 8b). Such distribution illustrates the extensive land use change experienced in south-central Chile over the last 50 years, where native forests have been progressively converted into agricultural lands and forest plantations (Armesto et al., 2010; Miranda et al., 2015). This conversion has had significant impacts in forest ecosystem services such as water
provision (Jones et al., 2017; Lara et al., 2009).

Figures 8c and 8d show that the Far and Near North Zones have more homogeneous land cover types, with shrublands and impervious lands occupying more than 60 % of the catchment areas. In southern areas, the coverage of the dominat classes decreases substantially, transitioning towards a mosaic of different land cover types. Missing land cover data is presented in Fig. 8e, which should be accounted for if the land cover attributes of the affected catchments (i.e., the ones with portions in
Argentina, as shown in Fig. 2) are used for applications such as catchment classification or parameter regionalisation.

Because of the glaciological contributions to the water balance within the domain (Mernild et al., 2017; Le Quesne et al., 2009), we added two attributes (Table 3) based on information from the glaciers inventory described in Sect. 3.1.4. We found that 255 catchments (48 % of the total) have some degree of glacierisation, reaching up to 62 % in the Geike River catchment in the Southern Patagonia. The catchments with the largest degree of glacierisation (more than 15 %) are located in the Austral
and Southern Patagonia regions, followed by the Olivares and Volcan river catchments (about 14 %) in the Central Zone. The glaciers included in CAMELS-CL span 7321 km$^2$, corresponding to almost a quarter of the glacierised area in the Southern Andes (RGI Consortium, 2017).

### 3.2.4 Climatic indices

To allow direct comparisons between CAMELS (A17) and CAMELS-CL, climatic indices were computed for the same period
as in A17, i.e., water years 1990 to 2009, corresponding to 1 April 1990 to 31 March 2010 for Chile. If these indices are required for different periods, the formulae provided in the references from Table 3 can be used with the raw hydro-meteorological time series (available from CAMELS-CL website). The complete spatial and temporal coverages of meteorological variables allow the estimation of climatic indices for all 516 catchments – in contrast to hydrological signatures,

computed for a subset of catchments (Sect. 3.2.5). Precipitation and PET-based attributes were calculated for all precipitation products (Sect. 3.1.6), using the daily PET product (pet$_{har}$, Sect. 3.1.8).

The climatic attributes presented in Fig. 4 and 9 reveal basic features of Chilean climatology, described in more detail by Miller (1976) and Garreaud et al. (2017), among others. Mean annual precipitation ranges from less than 10 mm yr$^{-1}$ in the

Atacama Desert (northern Chile) to more than 3000 mm yr$^{-1}$ in western Patagonia (Fig. 4b). Such a marked precipitation gradient reflects the relative influence of the subtropical, semi-permanent Southeast Pacific anticyclone, and the frequent incursion of frontal systems at higher latitudes. The frequency of high precipitation events also increases southward, with a maximum in south-central Chile (Fig. 9d). The Andean domain in the Far North (Chilean Altiplano), receiving about 300 mm yr$^{-1}$ above 4000 m a.s.l., is influenced by the monsoonal regime developing over the interior of the continent. In addition to

the N-S gradient, precipitation increases strongly in the west-east direction due to the orographic enhancement of air masses over the windward slope of the Andes Cordillera (a factor of 2-3 between lowlands and windward slopes; Viale and Garreaud, 2014). PET has a more restricted range than precipitation (400-1400 mm yr$^{-1}$; Fig. 4d), therefore, the aridity index (PET/P, Fig. 9c) is higher in northern Chile (> 1.0) compared to that of southern regions (< 1.0). A positive precipitation seasonality (Fig. 9a) in northern Chile indicates precipitation peaks during summer (DJF), following the monsoonal precipitation regime

governing in this region (Fig. 9f). In contrast, the negative seasonality values obtained for all macro-zones, except the Far North and Southern Patagonia, illustrate the increased storm frequency and high precipitation events in most of the country during the winter (JJA) (Fig. 9a). Seasonality values close to zero indicate uniform precipitation throughout the year in Southern Patagonia (Fig. 9a and 9f). The zero-temperature isotherm during winter storms ranges between 1500 and 4000 m a.s.l., so most of the precipitation is liquid along the coast and interior valleys (Fig. 9b), while a larger fraction of solid

precipitation is obtained in high-elevation basins.

### 3.2.5 Hydrological signatures

Hydrological signatures were computed for the period 1 April 1990 to 31 March 2010, as in Sect. 3.2.4. To exclude the effects of anthropic intervention on hydrologic response, we selected 94 catchments with valid daily streamflow records in at least 85 % of the period, based on the following criteria: interv_degree lower than 0.1 (i.e., less than 10 % of the annual streamflow

allocated to surface rights), large_dam equal to zero (absence of large dams within the catchment), imp_frac lower than 5 % (negligible urban areas), copr_frac lower than 20 % (negligible irrigation effects) and fp_frac lower than 20 % (negligible forest plantations effects). Further, we excluded glacier dominated catchments by selecting glacier_frac lower than 5 %. It should be noted that, despite of calculating hydrological indices for a subset of catchments, raw daily time series for all 516 catchments are included in CAMELS-CL database. These time series and the formulae provided in Table 3 may be used if the

signatures are required for different time periods.

Figure 10 illustrates the spatial distribution of 12 (out of 14) hydrological signatures (Table 3), revealing the leading patterns of catchment responses. Both mean daily flow and runoff ratio increase from the Far North to the Southern Zone, showing strong correlations with mean annual precipitation (Fig. 4b) and the aridity index (Fig. 9c). Further, runoff ratio shows a

positive west to east gradient (i.e., increase towards the Andes), reaching values above one in mountain catchments in the Southern Zone. These non-behavioural catchments are further analysed in Sect. 4.1. Mean half-flow dates (Fig.10c) present a similar west to east gradient, with higher values in steep (Fig. 6c) snow-dominated (Fig. 9b) basins in Central Chile – where the most frequent season for low precipitation days is DJF (Fig. 9i).

The mid-segment slope of the flow duration curve (FDC, Fig. 10d) – a signature that quantifies flashiness of runoff – shows that slow basin-averaged responses occur in the Far North and part of the Near North, in spatial correspondence with high baseflow index (Fig. 10e) and low discharge precipitation elasticity (Fig. 10f). Such behaviour is expected in this region due to substantial subsurface and groundwater contributions to total runoff. Although flashiness of runoff and discharge elasticity to precipitation (baseflow index) are relatively higher (lower) and show some correlation towards the south, no clear spatial

gradients are observed within the domain spanning from Central Chile to Southern Patagonia. Fig. 10f shows negative elasticity values (-0.13 and -0.03) in two catchments located in Austral Zone and Southern Patagonia. Such values indicate a negative annual runoff anomaly in response to positive annual precipitation anomaly, which is not expected in near natural catchments. We attribute this behaviour to two main factors. First, there is a numerical problem in the formula used to calculate streamflow elasticity (Eq. 7 in Sankarasubramanian et al. (2001), adapted in A17, Table 3) when annual precipitation of a single year

approaches the long-term mean, causing the elasticity to approach infinity (Sankarasubramanian et al., 2001). A second factor is the use of incomplete streamflow daily records. Since the elasticity is computed from concurrent daily streamflow and precipitation, its calculation in catchments with missing streamflow records can be problematic. This can be particularly important in snow dominated catchments (delayed runoff response to precipitation) and in catchments with a weak precipitation seasonality (i.e., precipitation falling during the whole year, Fig. 9a close to zero), which is the case for catchments

in the Austral Zone and Southern Patagonia.

The examination of signatures related to extreme (high or low) streamflow conditions exposes some interesting features. Although no clear spatial relationship is observed between high flow signatures (Fig. 10g-i), similar spatial distributions of low flow frequencies (Fig. 10j) and mean low flow durations (Fig. 10k) are obtained across the country. Q95 (Fig. 10i) and Q5 (Fig. 10l) provide generally similar patterns to those of mean daily discharge (Fig. 10a), with positive increases from the

Far North to the Southern Zones, and a positive west to east gradient. The comparison between the signatures displayed in Fig. 10g-l and climatic indices in Fig. 9d-i highlight the complex relationship between climate and hydrologic catchment behaviour. For example, the spatial structure in the frequency of low/high precipitation days is not reflected in the spatial distribution of high/low flow frequencies. A similar disjunction is observed between the duration of low precipitation (Fig. 9h) and low flow (Fig. 10k) events, whereas those catchments with low duration of high precipitation events also provide low durations in high

flow events.

Sharp variations in hydrological signatures (Fig. 10) – in contrast to generally smooth patterns in climate indices (Fig. 9) – are the result of complex, nonlinear processes across a range of spatiotemporal scales, enhanced by heterogeneities in topography, soils, vegetation, geology and other landscape properties. Careful attention should be paid to such interactions and to the uncertainties involved in the calculation of hydrological signatures, in particular when attempting to extrapolate hydrological

behaviour from gauged to ungauged basins based on climatic similarities alone (Westerberg et al., 2016; Westerberg and McMillan, 2015).

### 3.2.6 Human intervention

Figure 11 summarises water rights records used to characterise human intervention degree within the catchments. We can see
that the number of surface rights (Fig. 11a) increases from north to south, while the number of groundwater rights (Fig. 11d) increases from east to west. Although these values do not provide information about allocated volumes, they show how many water rights holders interact to coordinate the water use within a particular catchment. CAMELS-CL database provides information about each water right within a catchment (not only the attributes with synthetized information), in case more detailed analyses are required.

In terms of allocated surface and groundwater flows (Fig. 11b and Fig. 11e, respectively), we only considered consumptive permanent water rights. Further, we considered only water rights recorded as volume per time, since water rights expressed as "shares" (6 % of the national water rights database) were not provided with their corresponding conversion into volume units (DGA, 2016b). It should be noted that shares rights are the oldest (allocated prior to the 2005 water code reform), thus probably representing a majority of the rights within the Central Zone (region that concentrates the oldest rights).

The above limitations may lead to an underestimation of the allocated flow due to (at least) the following reasons: (i) non-consumptive rights may have their restitution points outside the catchment boundaries (however, they were not considered for the allocated flow calculation); (ii) shares rights are disregarded; (iii) there is missing information, and therefore some rights may be omitted (Sect. 3.1.10). On the other hand, allocation estimates may differ considerably from the actual extraction within a catchment. Possible reasons for this are the sub/over use of a granted allocated flow and unauthorised extractions of surface
and groundwater.

Despite the limitations of the water use dataset and the attributes presented in Fig. 11, water rights information is critical to quantify human intervention, and it has not been officially processed at the catchment scale in Chile. To quantify the intervention degree within a catchment, we calculated the interv_degree attribute (described in Table 3 and illustrated in Fig. 11c) as the ratio between the annual surface flow allocated within a catchment, and the catchment mean annual runoff. This
attribute indicates how much of the annual runoff generated – in average – within a catchment, corresponds to the water volume allocated as consumptive surface rights. Further, we defined a binary attribute to characterise the presence of reservoirs within a catchment (large_dam in Table 3), using 0 if there are no dams, and 1 if there is at least one dam. A limitation of these human intervention attributes (interv_degree and large_dam) that should be considered for hydrological applications is that they do not incorporate information about groundwater extractions.

To quantify the urbanised fraction of a catchment – another important factor modulating catchment response –, we used the impervious fractional area attribute (imp_frac in Table 3), which usually contains urban areas. However, this land cover type is the worst classified class, since urban areas have mixed pixels of vegetation and paved surfaces (Zhao et al., 2016). The

urban fraction of the catchments (assumed to be equal than imp_frac) varied between 0 % and 7 % for most catchments (only one catchment had imp_frac = 25 %).

## 4 Uncertainty in precipitation and PET

### 4.1 Precipitation assessment

To assess precipitation uncertainty, we looked at the inter-product differences across the study domain. To this end, we defined a precipitation spread attribute (p_mean_spread, Table 3) as the standard deviation of basin-averaged mean annual precipitation from the four different products, normalised by multi-product mean. To allow such inter-comparison, we used data from the concurrent period 1998-2014 (Table 2), and excluded catchments located southern than 50° S (since CHIRPS and TMPA cover up to 50° S). Given the different nature of the assessed precipitation products, the spread attribute can be interpreted as a
measure of precipitation uncertainty. The underlying assumption is that similar values from different data sources indicate regions with higher confidence in precipitation estimates.

Figure 12 displays catchment-scale mean precipitation and the precipitation spread index for three macro-regions: North (northern than 34° S), which includes the Far North and Near North macro-zones; Central-South (between 34° S and 43° S); and Austral-Patagonia (southern than 43° S). Mean precipitation estimates (p_mean) have a larger spread in the North (Fig.
12a), indicating larger uncertainties in this domain. We attribute these higher relative errors to methodological challenges for detecting events and estimating their intensities in this arid sub-domain, where the occurrence of precipitation events is relatively rare (note the different scale used for p_mean in Fig. 12d). By contrast, considerably larger precipitation amounts (Fig. 12e-f) and lower spread values (Fig. 12b-c) are obtained in Central-South and Austral-Patagonia, which is expected given the relation of p_mean_spread to precipitation mean values. On the other hand, if we look at absolute inter-product differences
(Fig. 12d-f), the Central-South region features the largest standard deviation in basin-averaged mean annual precipitation from the four different products (median value of 0.80 mm $d^{-1}$, compared to 0.29 mm $d^{-1}$ and 0.51 mm $d^{-1}$ in the North and Austral-Patagonia regions, respectively). This is expected given the larger mean annual values over this humid region.

Although the effects of large precipitation uncertainty on streamflow modelling in the North are not straightforward to determine, some insights can be gained from our analyses. First, surface runoff is not very sensitive to variations in
precipitation (i.e., small runoff elasticity values in Fig. 10f), suggesting a weak propagation of precipitation errors by hydrological models. Second, groundwater has the largest contribution to streamflow in this domain (largest baseflow indices in Fig. 10e and sedimentary rocks as the most common geologic class illustrated in Fig. 7a), especially in the presence of Andean peatlands (represented by the wet_frac land cover attribute). This highlights the need to pursue a realistic representation of groundwater mechanisms in numerical models. Additionally, aquifer boundaries may be quite different from
surface catchment boundaries, and therefore accurate delineations are needed to ensure a good representation of surface-groundwater interactions (e.g., Sar et al. 2015; Arkoprovo et al. 2012; Ivkovic et al. 2009).

The ensemble spread of precipitation estimates is a measure of disagreement among the various products rather than a measure of accuracy, which should be quantified using ground observations (e.g., Zambrano-Bigiarini et al. 2017). Such analysis is beyond the scope of this paper, since the assessment of different precipitation products at the basin scale is typically conducted by forcing one or more hydrological models with the different precipitation datasets over the selected study area (e.g., Bisselink

et al. 2016; Thiemig et al. 2013; Su et al. 2008).

As an alternative to the model-based approach, we examined the consistency of catchment precipitation estimates based on long-term runoff ratios in 94 near-natural catchments (Fig. 12g-i), selected following the criteria presented in Sect. 3.2.5. Although there are large inter-product differences in runoff ratios in the North (consistent with large p_mean_spread values in Fig. 12a), relatively low runoff ratio values (<0.4) are obtained, as expected given the arid and semi-arid conditions in this

region. By contrast, there are catchments with runoff ratios larger than 1 in Central-South and Austral-Patagonia, indicating that there is more water leaving the catchment than the total amount entering as precipitation. Assuming that streamflow data and catchment area are reliable, and that changes in storage and groundwater contributions are negligible, such cases indicate precipitation underestimation by the various products. In the Central-South (Austral-Patagonia) region, the MSWEP (CR2MET) dataset provides 8 % (20 %) of catchments runoff ratio > 1 – i.e., the smallest among all products. In both domains,

the TMPA dataset provides the largest fraction of catchments with runoff ratio > 1 (54 % in Central-South and 70 % in Austral-Patagonia). Such underestimation of TMPA, as well as other satellite precipitation estimates, was also reported by Hobouchian et al. (2017) and Zambrano-Bigiarini et al. (2017).

To further explore differences and systematic biases in within the assessed products, we used the Budyko framework (Budyko, 1971) to diagnose the factors affecting the quality of the precipitation datasets. This framework links climate to catchment

runoff and evapotranspiration in a simple and easy to interpret visualisation. Figure 13 shows the evaporative index (EI, the ratio of mean annual evapotranspiration to the mean annual precipitation), estimated as one minus the runoff ratio (i.e., assuming that changes in storage and groundwater contributions are negligible; Sposito, 2017), as a function of the aridity index for the 94 near-natural catchments over the period 1998-2014. Figure 13 illustrates how the evapotranspiration and runoff rates within this highly diverse set of catchments are governed by the available energy and precipitation, e.g., – for a

given amount of precipitation – runoff exceeds evapotranspiration when the available energy and PET are relatively low (points below the energy limited line in humid regions).

Negative EI values in Fig. 13 represent non-behavioural combinations of precipitation and runoff (Berghuijs et al., 2017). Under the assumption that precipitation estimates represent a relatively larger source of uncertainty compared to runoff, all the points with EI < 0 indicate those catchments where mean annual precipitation is underestimated (i.e., runoff ratios > 1).

Therefore, Fig. 13a-d indicate that all precipitation products systematically fail in humid catchments with steep topography (slopes greater than 150 m km$^{-1}$), in agreement with the limitations reported for different satellite precipitation products over the same domain (Hobouchian et al., 2017). The systematic precipitation underestimation can be attributed to the complex topography of headwater catchments and the scarcity of ground stations at high elevations. In fact, 90 % of the 500 rain gauges located south of 34° S are placed below 1000 m a.s.l. These precipitation errors pose challenges for hydrological applications

and the best strategy to address this will depend on the application. It should be noted that 57 catchment attributes (out of 70) and six hydro-meteorological time series (streamflow, min, max and mean temperature, PET, and SWE) do not rely on precipitation. Therefore, some applications may not need to directly address this limitation (e.g., catchment characterisation, water allocation analysis, classification studies not using similarity of climatic indices). Nevertheless, other applications (e.g.,

hydrological modelling) will have to deal with the uncertainties of this forcing variable. There are strategies for this, including to set up a parameter calibration scheme flexible enough to compensate errors in meteorological input data (Elsner et al., 2014), or to try to correct precipitation estimates based on streamflow (Henn et al., 2015) and snow observations (Henn et al., 2016).

### 4.2 PET assessment

To assess the quality of the PET products described in Sect. 3.1.8, we used a different approach than in Sect. 4.1, since a basin-

scale PET estimation cannot be evaluated based on observed streamflow. In this case, the evaluation was made with an independent set of PET data, calculated with daily observations for the period 2010-2014 from 75 meteorological stations maintained by the Chilean National Institute of Agricultural Research (INIA, 2017). For each site, we calculated two PET time series: (i) a daily time series obtained through the Hargreaves formulae (Hargreaves and Allen, 2003) fed with INIA temperature observations, called $INIA_{har}$ hereafter,  and (ii) an 8 day accumulated time series based on the FAO Penman-

Monteith reference crop evapotranspiration (Allen et al., 1998), called $INIA_{ET0}$. $INIA_{har}$ and $INIA_{ET0}$ were used to evaluate the corresponding pixels of $PET_{har}$ and $PET_{mod}$, respectively.

The evaluation metrics used in these comparisons, spatially averaged within the macro-zones, are summarised in Table 4. These results indicate good agreement between $PET_{har}$ and $INIA_{har}$, with correlation coefficients (r) greater than 0.92 throughout the national territory, except in the Far-North macro-zone, where we found a weaker correlation (r = 0.76). The

ratios between mean $PET_{har}$ and mean $INIA_{har}$ indicate that $PET_{har}$ underestimates (overestimates) up to 8 % the observation-based PET in the Far and Near North arid regions (the Southern and Austral Zones humid regions).

The comparison between $PET_{mod}$ and the $INIA_{ET0}$ led to r values greater than 0.80 within the domain, excepting the Far North, where the correlation was below 0.20 in all available stations. The ratios between $PET_{mod}$ and $INIA_{ET0}$ means indicate that the first one systematically overestimates station-based calculations, which was also found by Westerhoff, (2015). Such systematic

biases may be explained by the theoretical differences between $INIA_{ET0}$ and $PET_{mod}$ calculated in MOD16. $INIA_{ET0}$ represents a potential condition for a regular crop height of 0.12 m and a fixed surface resistance and albedo, which is not the case for $PET_{mod}$, which includes a more complete parameterisation of those variables based on vegetation characteristics. Further, $INIA_{ET0}$ uses local meteorological observations, while $PET_{mod}$ uses global sources that may not capture meteorological variations at the local scale. If an application requires it (e.g., irrigation or hydrological modelling applications), the biases

reported $PET_{mod}$ biases can be corrected with conventional statistical methods (e.g., Maraun and Widmann, 2018). Other spatio-temporal analyses (e.g., drought monitoring) may directly apply $PET_{mod}$ due to its high correlation with ground $ET_0$ estimates.

Since INIA records were used differently for evaluating $PET_{har}$ and $PET_{mod}$, a direct comparison between both assessments is not possible, although they provide valuable information about the quality of PET products across the territory. Furthermore, the formulation behind the two gridded products have different trade-offs. $PET_{mod}$ is based on the Penman-Monteith equation that solves the surface energy balance, including parameters such as albedo and FPAR/LAI, whereas $PET_{har}$ is calculated from

an empirical approach based only on air temperature. $PET_{har}$ has a coarser spatial resolution ($5\times5$ km$^2$) compared to $PET_{mod}$ ($1\times1$ km$^2$), which may induce to larger errors over complex topography (e.g., mountain catchments) due to the local variations in potential evapotranspiration with changes in slope and aspect. On the other hand, $PET_{har}$ covers a longer period (1979-2016, the same as $T_{min}$ and $T_{max}$ from Sect. 3.1.7) compared to $PET_{mod}$ (2000-2014), which is more suitable to characterise climatic trends.

**5 Impacts of human intervention on catchment behaviour**

Large sample hydrology is a suitable framework to explore anthropic impacts on catchment behaviour through comparative analysis over a broad range of hydroclimatic conditions and catchment characteristics. Such assessment is critical when addressing the question of how climate change will affect global water supply (Vörösmarty et al., 2007). However, it remains unsolved how to generalise the results from different studies. For example, Poff et al. (2006) examined the effects of land use

on hydrological regimes (e.g., peak and low flows, runoff variability) in 158 basins within the CONUS, finding region-dependent changes in specific metrics. Ochoa-Tocachi et al. (2016) analysed the impacts of land use on the hydrology of 25 Andean catchments, finding that anthropogenic influences propagate towards increased streamflow variability and decreased catchment regulation capacity and water yield. More recently, Tijdeman et al. (2018) examined the effects of human intervention on streamflow drought characteristics across 187 catchments in England and Wales, concluding that most human-

influenced catchments did not have drought characteristics different from those expected for near-natural conditions.

In this work, we used hydrological signatures to describe catchment behaviour and the interv_degree attribute (Sect. 3.2.6) to characterise the level of human intervention. Figure 14 (panels a-d) display scatter plots between four hydrological signatures and the logarithm of the intervention degree index – which accounts for consumptive, continuous surface water rights. Different colours indicate the aridity of each catchment, which is a major driver of hydrological behaviour (as showed in Fig. 13). These

plots show that larger human intervention is associated to decreased annual flows and runoff ratios, especially in drier catchments. Interestingly, a larger number of consumptive surface rights (larger interv_degree values) is reflected on decreased elasticity of runoff with respect to precipitation, supported by low p-values. Since these scatter plots do not allow to separate the effects of aridity and human intervention, we binned the data and used boxplots to disentangle such effects. Figures 14e-h show the boxplots with these hydrological signatures for the classified catchments, binned according to their aridity (humid:

aridity below 0.8, medium: aridity between 0.8 and 1.5, and arid: aridity above 1.5) and their degree of human intervention. Catchments with low (high) intervention were defined by the interv_degree values lower (greater) than 5 % and the large_dam attribute equal to zero (one).

The dispersion in hydrological signatures among wet and medium catchments is large, and no significant difference is found between catchments with high and low human intervention. In contrast, dry catchments (zoomed view in Fig. 14i-l) reveal differences in hydrological signatures for high and low human intervention (i.e., median values in highly disturbed catchments are below the first quartile of low intervention catchments). In agreement with the scatter plots, the annual flows (Fig. 14i) and runoff ratios (Fig. 14j) decrease in catchments with larger number of consumptive surface rights, which is expected due to withdrawals of water within water-scarce regions. Further, highly disturbed arid catchments feature lower runoff sensitivities to precipitation compared to less disturbed ones (Fig. 14k), which could be attributed to altered runoff generation mechanisms associated to water withdrawals and reservoirs. Figure 14l shows that there is likely less variation in daily runoff – represented by the mid-segment slope of the flow duration curves – within highly disturbed arid catchments. The results found in arid catchments (i.e., water-scarce regions) provide new evidence of the potential impacts of human intervention on water supply. However, further research is needed to assess the causality of the correlations found here.

## 6 Concluding remarks

The CAMELS-CL dataset presented here provides novel information at the catchment scale in Chile, within a region that is largely underrepresented in large-sample studies. CAMELS-CL includes daily streamflow data and a suite of basin-averaged hydro-meteorological variables, including precipitation, temperature, potential evapotranspiration, and snow water equivalent for 516 catchments in the country. The dataset also includes shapefiles of drainage area boundaries related to streamflow gauge locations, overcoming the main limitations of the official national hydrographic network (DGA; CIREN, 2014). Further, we synthesised diverse and complementary datasets to compute 70 catchment attributes describing topography, geology, land cover, climate, hydrology, and anthropic intervention.

We described the advantages and main limitations of the datasets used to derive the various catchment attributes, which should be considered when using CAMELS-CL for selecting catchments and interpreting results. The main spatial patterns of catchment attributes and their inter-relationships were analysed across the entire domain (4300 km), which includes high altitude catchments and five different primary climatic regimes. The main conclusions of this analysis are as follows:

- the Andes position along the country largely explains the elevation and slope gradients from the 516 catchments,
- there is high geological variability between neighbouring catchments and within catchments,
- there is larger heterogeneity in land cover attributes towards southern regions,
- 48 % of catchments have some degree of glacierisation,
- the climatic attributes show a marked north to south precipitation gradient, combined with an orographic enhancement over the windward slope of the Andes Cordillera,
- hydrological signatures reflect the leading patterns of catchment hydrologic responses, with strong correlations between runoff (daily flows and runoff ratios) and mean annual precipitation and aridity index,

- there are substantial subsurface and groundwater contributions to total runoff in the northern regions (Far North and Near North),
- hydrological signatures feature, in general, sharper variations compared to patterns in climate indices, which is due to complex, process interactions across a range of spatiotemporal scales, enhanced by heterogeneities in topography, soils, vegetation, geology and other landscape properties, and data errors (as also observed in the USA by (Addor et al., in press).

The CAMELS-CL dataset was further used to assess hydro-meteorological biases and uncertainties in a large ensemble of watersheds in Chile, based on the comparison of various precipitation products – one national (CR2MET) and three widely used global products (CHIRPS, MSWEP and TMPA). Large discrepancies between products were detected in arid regions, which are explained by the methodological challenges associated with the rare occurrence of precipitation events in this region. Based on a water balance analysis using Budyko curves, we found systematic precipitation underestimation in headwater mountain catchments (high elevations and steep slopes) over humid regions. For these topographic characteristics and climatic conditions, all products failed to provide precipitation estimates that closed the water balance, with the TMPA product featuring the largest errors – in agreement with previous studies over the same domain (Hobouchian et al., 2017; Zambrano-Bigiarini et al., 2017). Such errors were attributed to the complex topography of headwater catchments and the scarcity of ground stations at high elevations (90 % of rain gauges located south of 34° S are placed below 1000 m a.s.l.). These limitations restrict our understanding of hydrological processes, posing challenges for streamflow modelling, water management and allocation. To alleviate these constraints, efforts should be put on improving the surface monitoring network at high elevations (> 1000 m a.s.l.). This would help to obtain more accurate remotely-sensed and model-based precipitation estimates in complex terrains. To deal with these precipitation errors in hydrological applications, we suggest to explore suitable strategies based on application requirements, and to use our findings to better interpret results. Further, we assessed the $PET_{har}$ and $PET_{mod}$ products with an independent observation-based PET. In general, both products showed good correlations with the observation-based PET for the complete domain, except in the Far North arid region. Regarding mean biases, $PET_{har}$ showed slight underestimation (overestimation) in the Far and Near North arid regions (the Southern and Austral Zones humid regions), compared with the observed-based PET. $PET_{mod}$ on the other hand, showed a systematic and larger overestimation of the observation-based PET within the complete domain, which was attributed to the theoretical differences between their formulations.

The assessment of precipitation and PET products showed different performances within the domain. Therefore, the choice of these products or similar datasets must be carefully made based on specific application and study requirements.

Finally, we used CAMELS-CL to explore the impact of human activities on catchment behaviour. We showed that larger human intervention is correlated with lower than normal annual flows, runoff ratios, elasticity of runoff with respect to precipitation, and flashiness of runoff, especially in drier catchments. These results not only illustrate how catchment behaviour can change with human intervention, but also reveal the potential of anthropic indices to explain shifts in hydrological systems.

In summary, this paper contributes to hydrological sciences by: (i) providing a unique dataset that can be used to advance our understanding of hydrological systems by learning from diversity, (ii) analysing the dominant spatial patterns of physical, climatic and hydrological attributes within the domain, (iii) assessing the quality of one national and three global precipitation datasets based on the observed water balance, (iv) assessing two PET products based on an independent set of PET point values calculated from meteorological records, and (v) examining the interplay between human intervention and changes in observed catchment response .

CAMELS-CL can be used to address research questions related to catchment classification, similarity and regionalisation, model parameter estimation, dominant controls on runoff generation, the impacts of different land cover types on catchment response, characterisation of drought history and projections, and climate change impacts on hydrological processes. CAMELS-CL will be continuously updated to incorporate new records and new datasets, which may include soils characteristics, water quality, seismology records, socio-economic indices and energy generation data. Additionally, new and more detailed information about the Chilean cryosphere will be included, complementing the global inventory processed here with national inventories of Chile and Argentina. Time series of streamflow, meteorological variables, and all catchment attributes described in this paper can be visualised from the CAMELS-CL explorer (http://camels.cr2.cl) and downloaded from https://doi.pangaea.de/10.1594/PANGAEA.894885.

**Data availability**

ASTER GDEM elevation data was downloaded from the NASA-JPL website (https://asterweb.jpl.nasa.gov/gdem.asp, accessed on December 2016). Geological data (GLiM) was downloaded from the PANGAEA Database (https://doi.pangaea.de/10.1594/PANGAEA.788537, accessed on May 2017). The land cover map was downloaded from Universidad de Chile website (http://www.gep.uchile.cl/Landcover_CHILE.html, accessed on May 2017). Glaciers inventory RGI 6.0 was downloaded from the Global Land Ice Measurements from Space initiative (https://www.glims.org/RGI, accessed on December 2017). The daily streamflow records were obtained from the CR2 website (http://www.cr2.cl/datos-de-caudales, accessed on March 2018). CR2MET precipitation and temperature products are were downloaded from the CR2 website (http://www.cr2.cl/datos-productos-grillados, accessed on March 2018). CHIRPS, TMPA and MSWEP precipitation products are publicly available and the datasets clipped within Chilean domain were downloaded from the CR2 website (http://www.cr2.cl/datos-precipitacion-satelital, accessed on March 2018). MODIS data was downloaded from the USGS website (https://e4ftl01.cr.usgs.gov/MOLT, accessed on May 2017). SWE data was provided by Gonzalo Cortes (Cortés et al., 2016). Water rights were downloaded from the DGA website (http://www.dga.cl/productosyservicios/derechos_historicos, accessed on May 2018). Dams location was downloaded from Ministerio de Bienes Nacional website (http://www.ide.cl/descarga/capas/item/embalses-2016.html, accessed on September 2017). The time series processed at the basin scale and the catchment attributes introduced in this paper are freely available from https://doi.pangaea.de/10.1594/PANGAEA.894885 and the CAMELS-CL explorer (http://camels.cr2.cl).

## Author contribution

This research was conceived by CAG, PAZ and JPB. CAG processed the catchment attributes, designed and wrote the manuscript with input from all co-authors. PMZ performed the analysis on hydrological signatures. JPB developed and processed the CR2MET products. JPB and RG performed the analysis on climatic indices. NA provided R codes for computing climatic indices and hydrological signatures, and performed the analysis on geological attributes. MG and CP performed the analysis on PET. MZB processed the satellite precipitation products. AL contributed with the analysis on land cover attributes. GC processed the SWE data. JM contributed ideas for analyses. AA processed the glaciers data. All authors have been involved in interpreting the results, discussing the findings, and editing the paper.

## Competing interests

The authors declare that they have no conflict of interest.

## Acknowledgements

This research emerged from the collaboration with many colleagues at the Center for Climate and Resilience Research (CR2, CONICYT/FONDAP/15110009). Camila Alvarez-Garreton is funded by FONDECYT Postdoctoral Grant N° 3170428. Pablo Mendoza received additional support from FONDECYT Postdoctoral Grant N° 3170079. Mauricio Zambrano-Bigiarini thanks FONDECYT 11150861 for financial support. The development of CR2MET was supported by the Chilean Water Directorate (DGA), through the National Water Balance Updating Project DGA-2319, and by the FONDECYT Grant N° 3150492. This study is a contribution to the Large-sample Hydrology working group of the Panta Rhei Research Initiative of the International Association of Hydrological Sciences (IAHS). We thank Guillermo Tapia from DGA for his advice on stream gauges location.

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

**Figures**

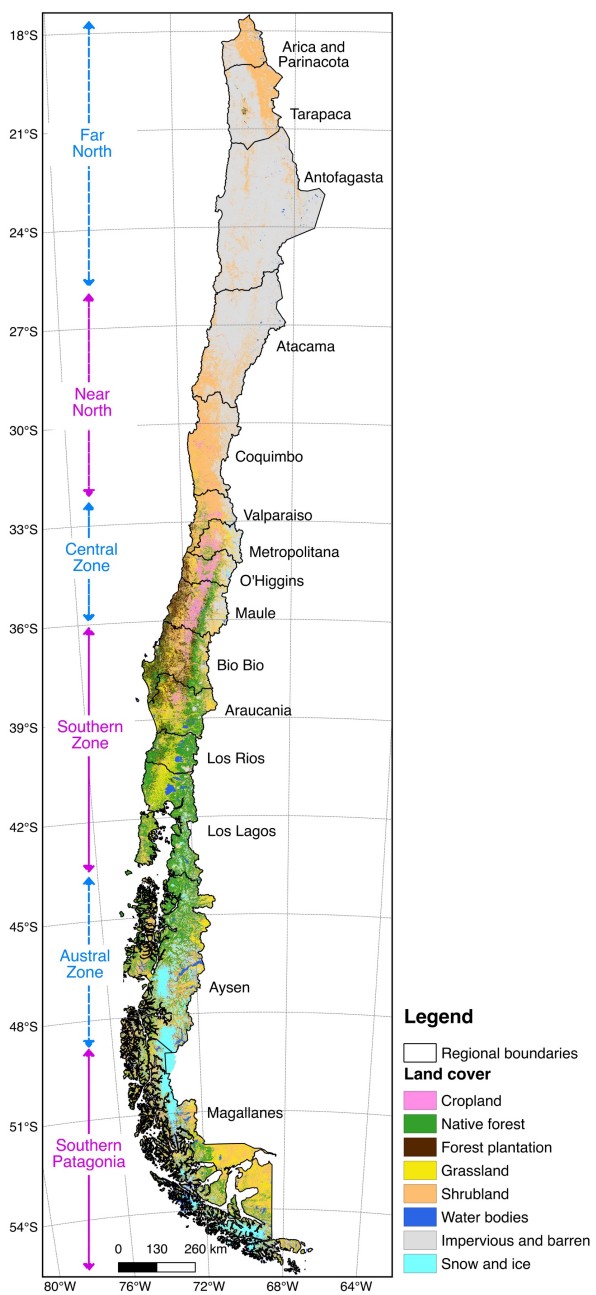

**Figure 1: Chilean regional boundaries and names, and the six defined macro-zones (blue and magenta arrows). The background colour correspond to the main land cover classes, obtained from Zhao et al. (2016).**

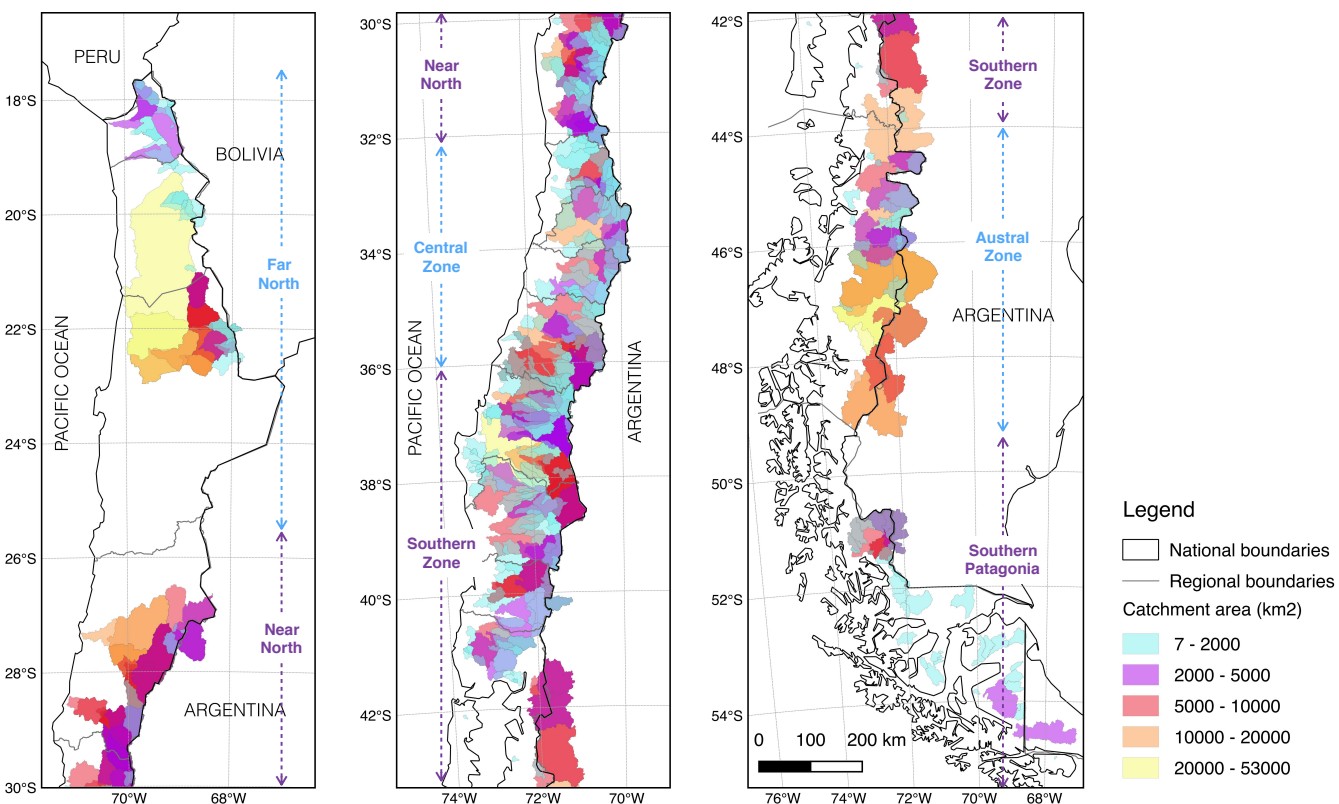

**Figure 2: Catchment boundaries and contributing areas (km²) of the 516 watersheds included in this study. The six defined macro-zones are indicated in blue and purple arrows.**

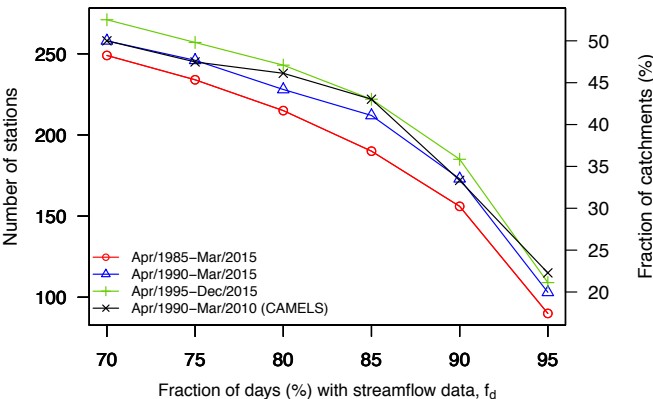

5     **Figure 3: Number of stations (left y-axis) having at least f_d % of days with daily streamflow records, for different periods. The right y-axis shows the percentage of catchments (out of 516) that meets the criterion. The period used in the CAMELS dataset (Addor et al., 2017) was included as a reference.**

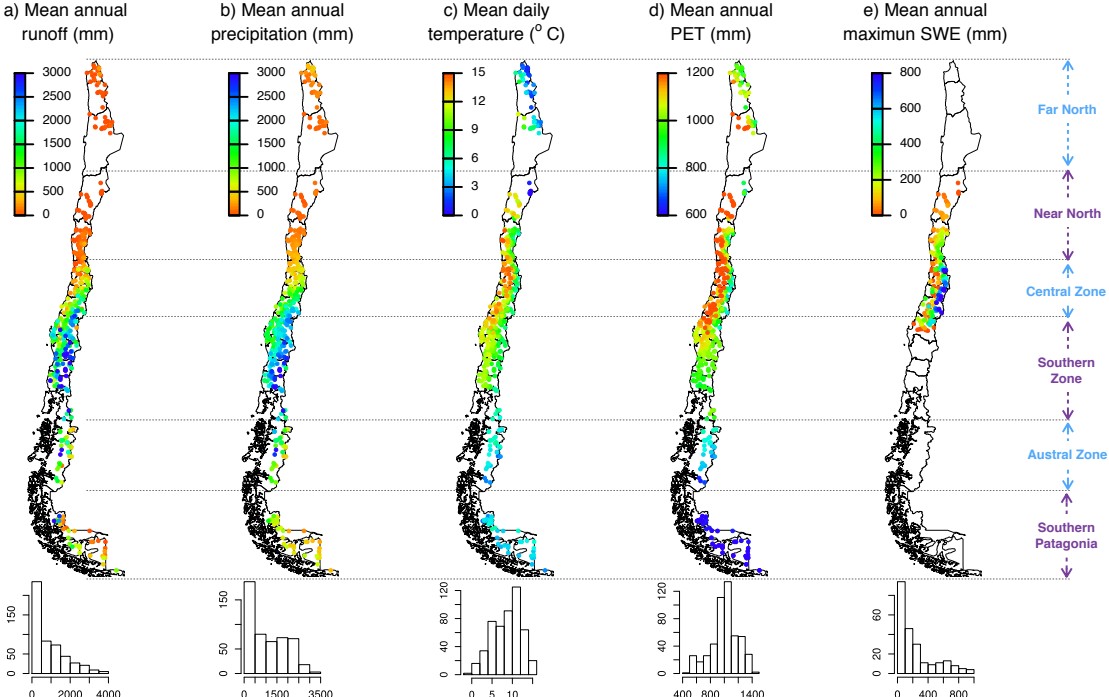

**Figure 4: Mean annual hydro-meteorological variables, calculated for the complete recording period of each variable. Panels b and d were generated with, precip$_{cr2met}$ and pet$_{har}$ products, respectively. The SWE product in panel e covers only the Near North and Central Zone. The histograms indicate the number of catchments (out of 516) in each bin. The points represent the location of catchment outlets.**

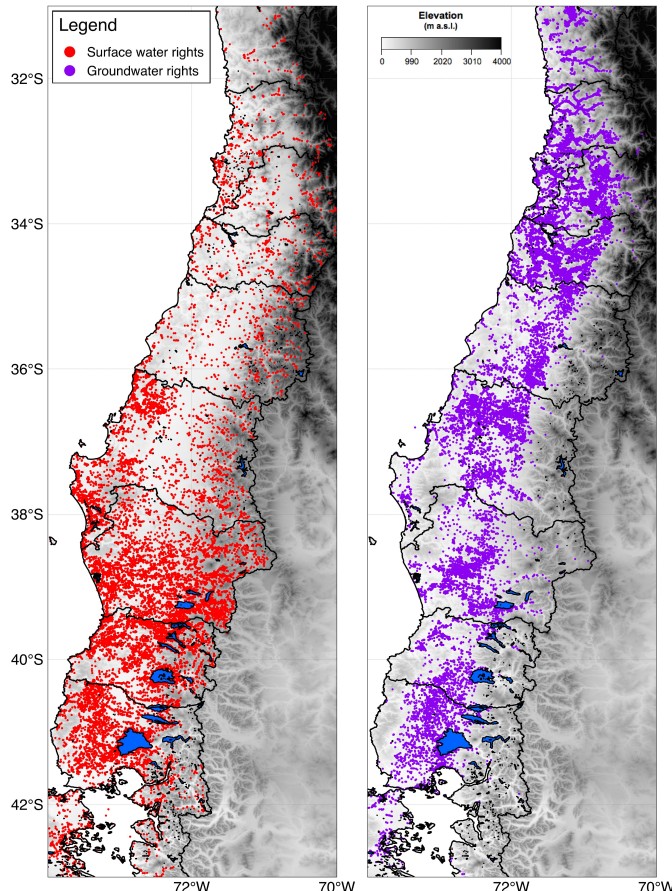

**Figure 5: Surface (left panel) and ground (right panel) water rights (consumptive, non-consumptive, permanent, eventual and alternated) granted by the Chilean Water Directorate (DGA) for a portion of the country. Background colours represent topography (greyscale) and the main water bodies (highlighted in blue).**

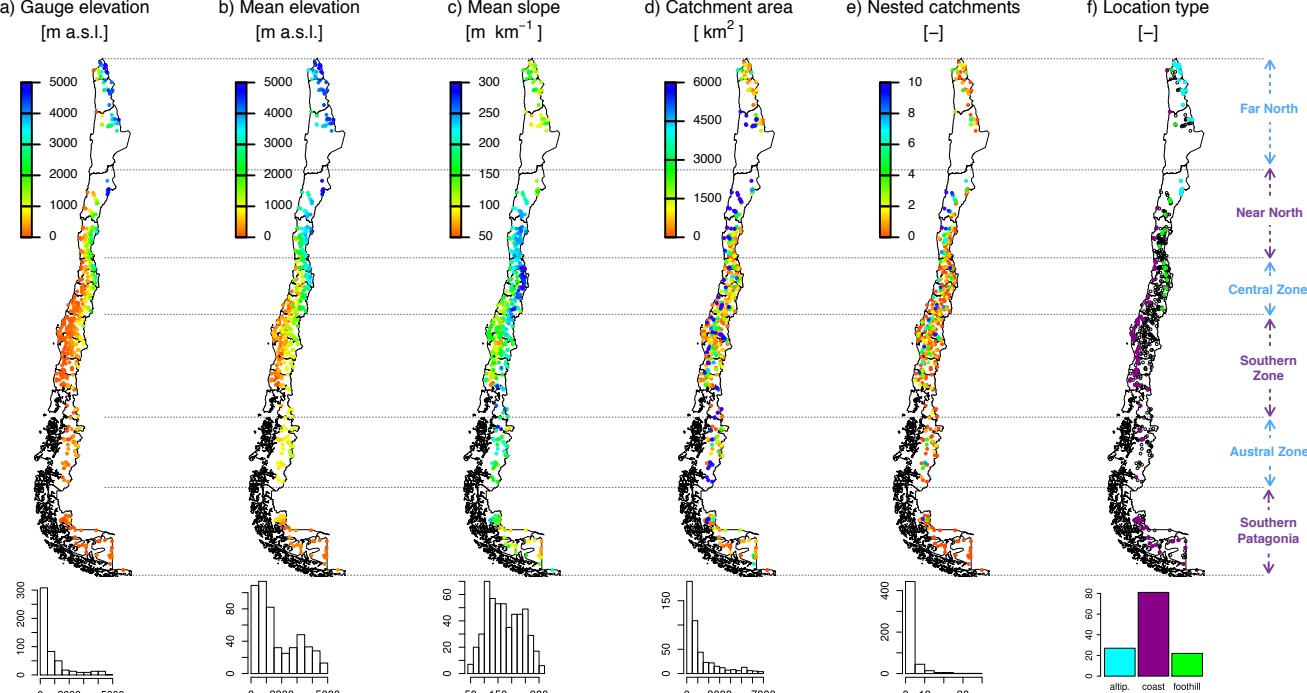

**Figure 6: Location and topography. For visualization purposes, catchment areas (panel d) are shown up to their 90<sup>th</sup> percentile. The histograms indicate the number of catchments (out of 516) in each bin.**

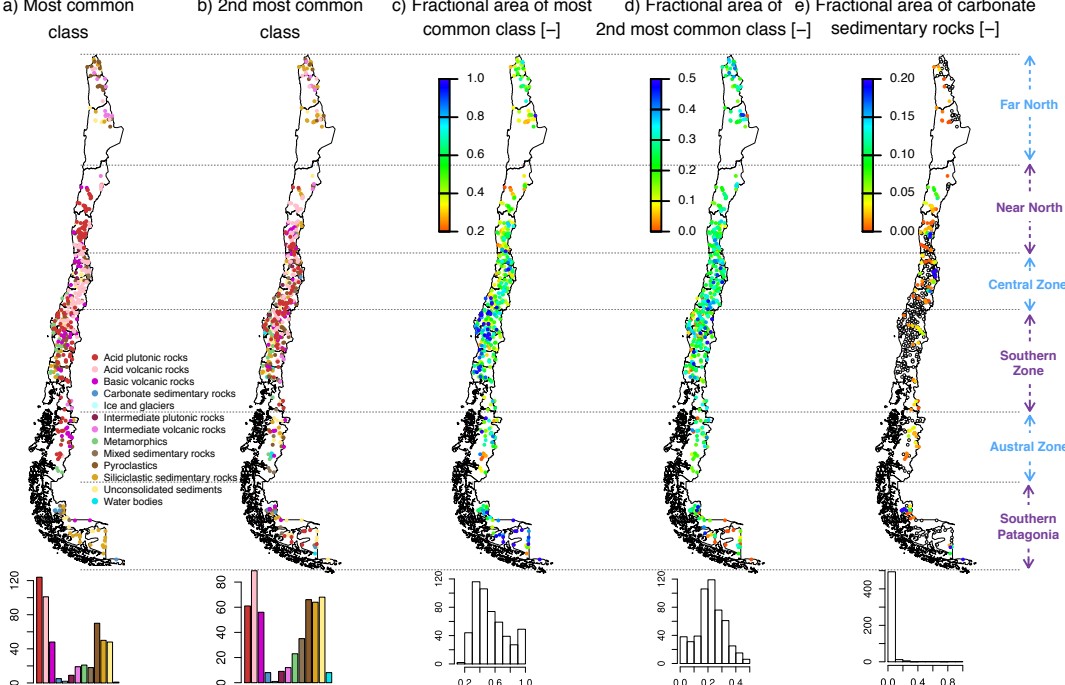

Figure 7: Geology attributes. The histograms indicate the number of catchments (out of 516) in each bin.

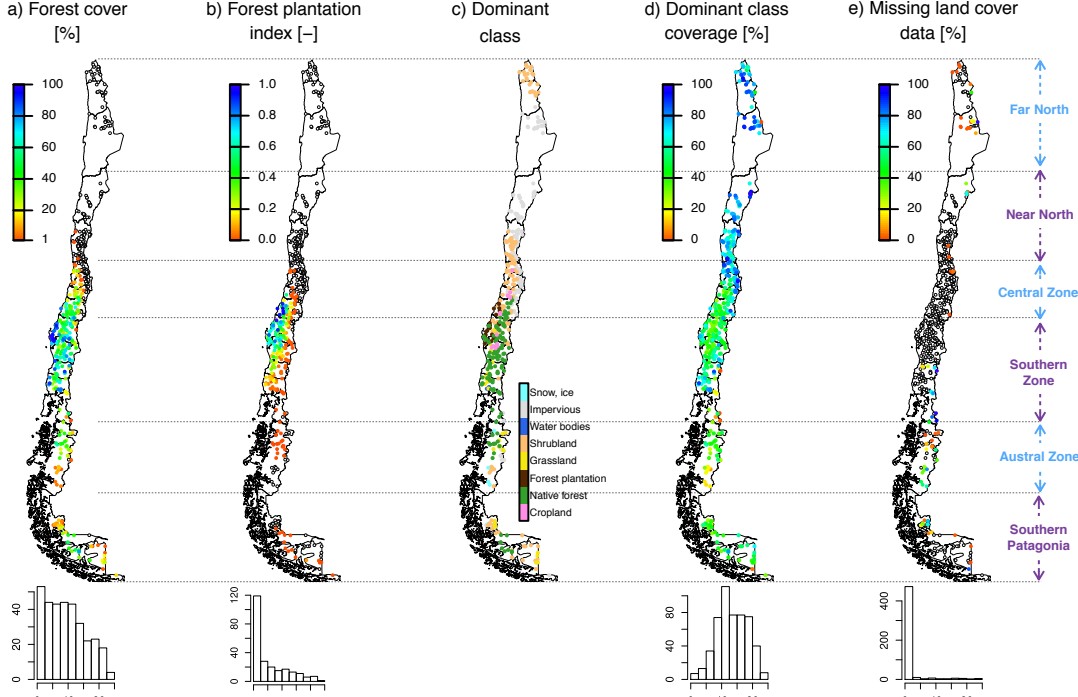

**Figure 8: Land cover characteristics. Values below colour bar lower limits are shown blank (i.e., blank points in panel a represent no forest cover, and in panel e represent no missing land cover data within those catchments). The histograms indicate the number of catchments (out of 516) in each bin.**

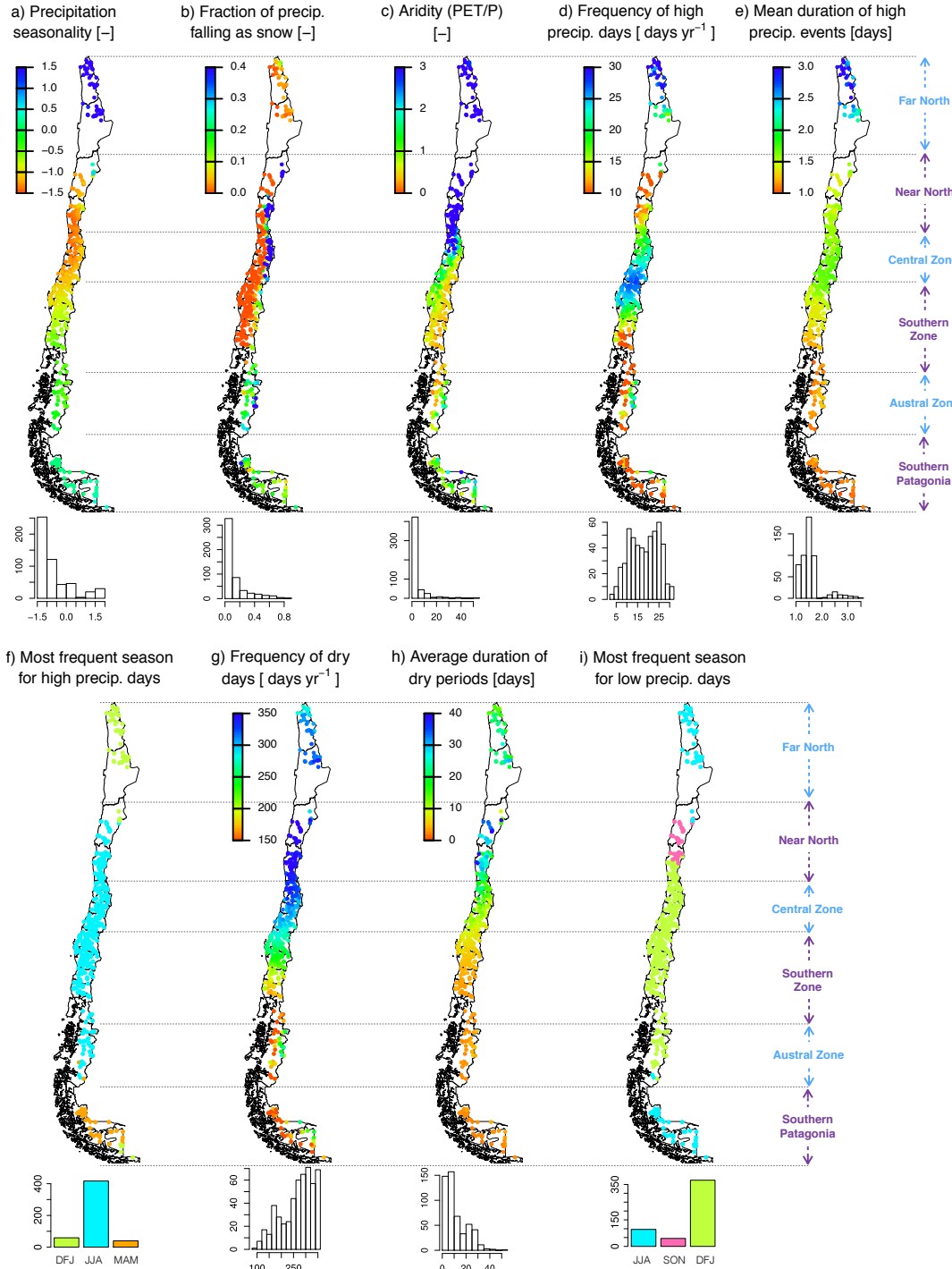

**Figure 9: Climatic indices (calculated from precip_cr2met product). The histograms indicate the number of catchments (out of 516) in each bin.**

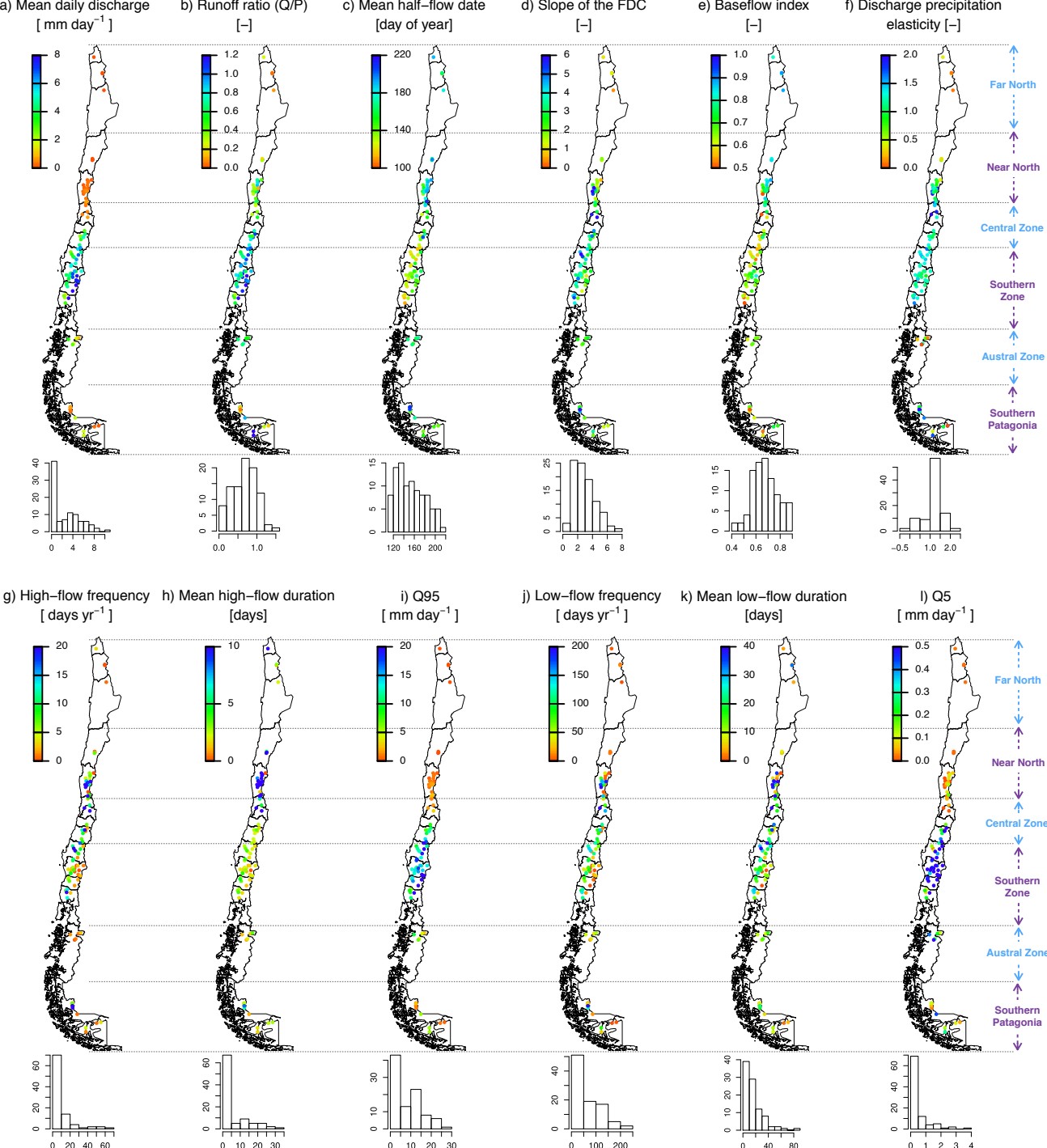

**Figure 10: Hydrologic signatures for 94 near-natural catchments. The histograms indicate the number of catchments (out of 94) in each bin.**

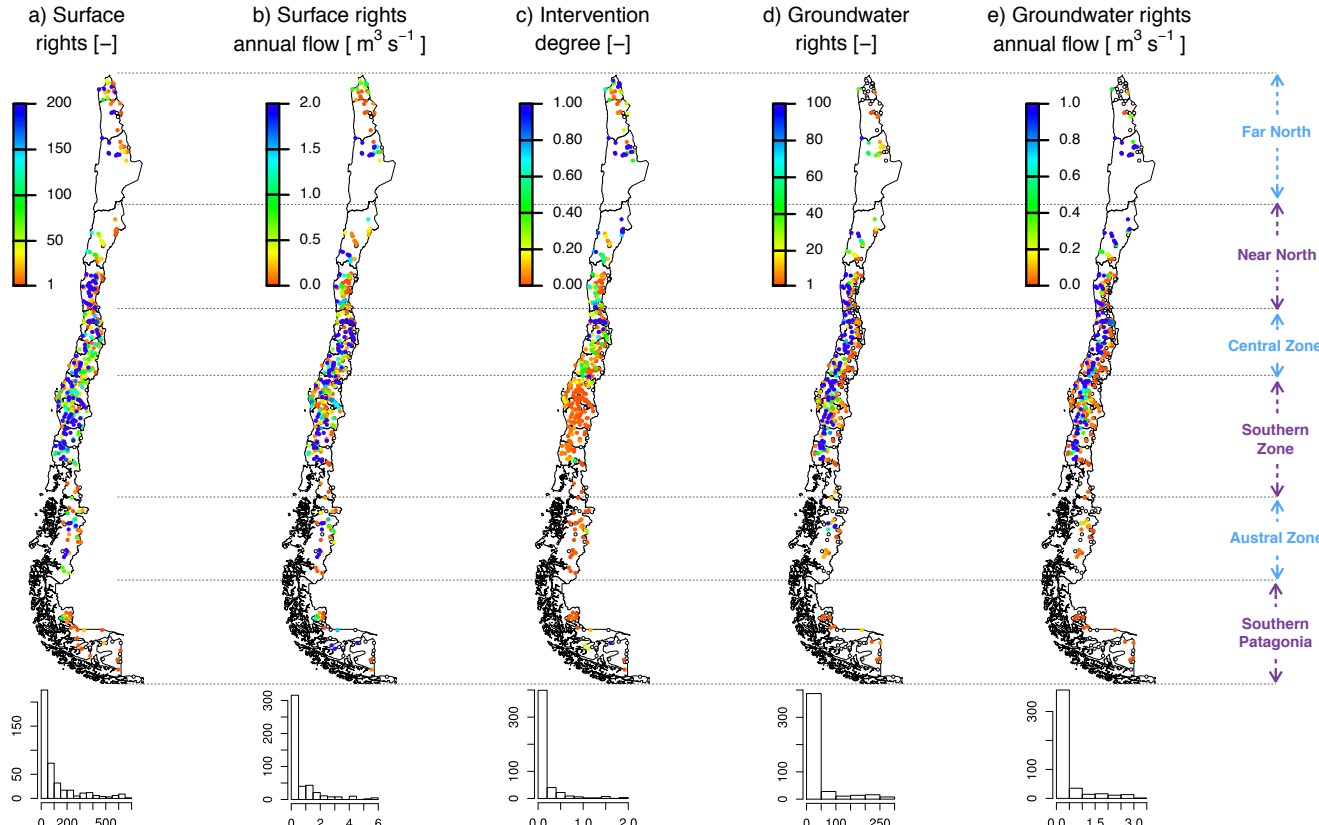

**Figure 11: Water rights attributes. For visualization purposes, the attributes in panels a, b, d and e are shown up to their 90th percentile. Attributes below the lower colour bar value are blank and above the upper colour bar value are blue. The histograms indicate the number of catchments (out of 516) in each bin.**

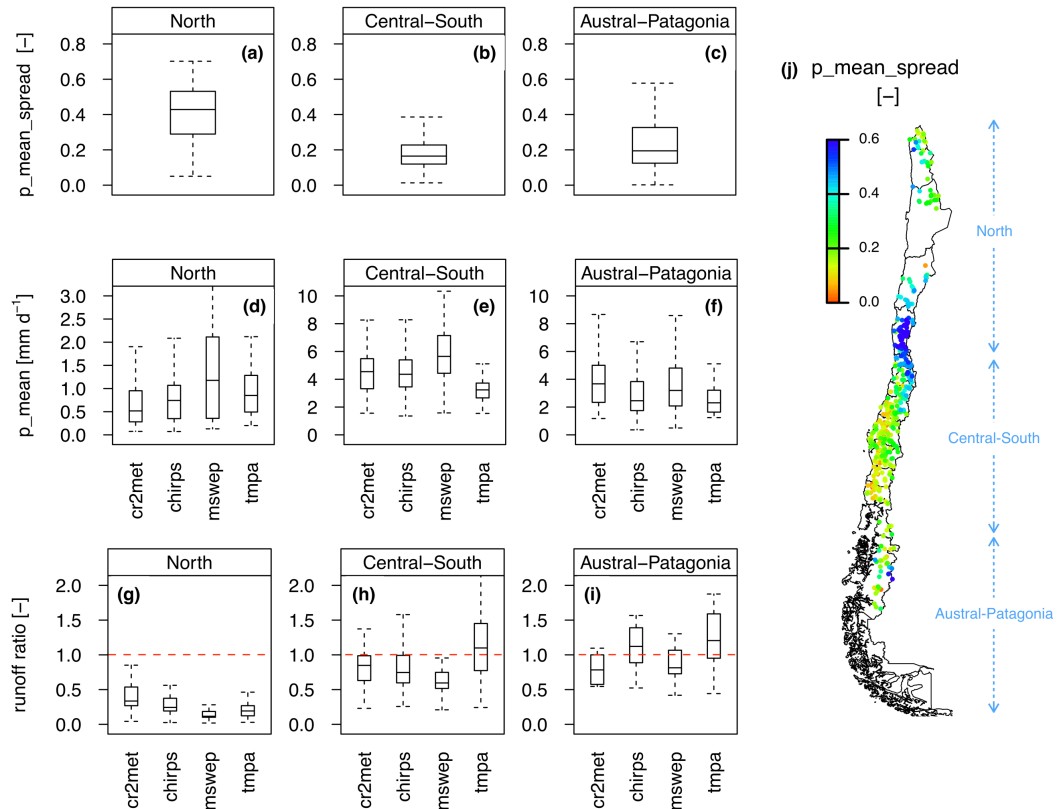

**Figure 12: Precipitation spread (p_mean_spread in panels a-c), mean annual precipitation (panels d-f), and runoff ratio (panels g-i), for the different precipitation products. The domain was divided in three main regions: North (northern than 34° S); Central-South; and Austral-Patagonia. Panel j shows the spatial distribution of p_mean_spread in these sub-regions.**

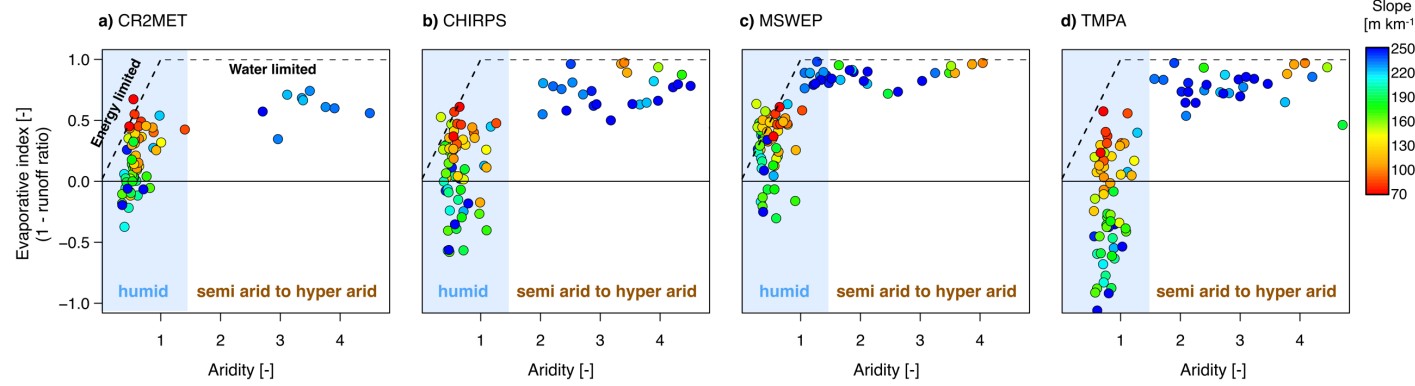

**Figure 13: Water balance for 94 near-natural catchments, illustrated in a Budyko scheme for CR2MET (panel a), CHIRPS (panel b), MSWEP (panel c) and TMPA (panel d). Markers are coloured by the catchment mean slope.**

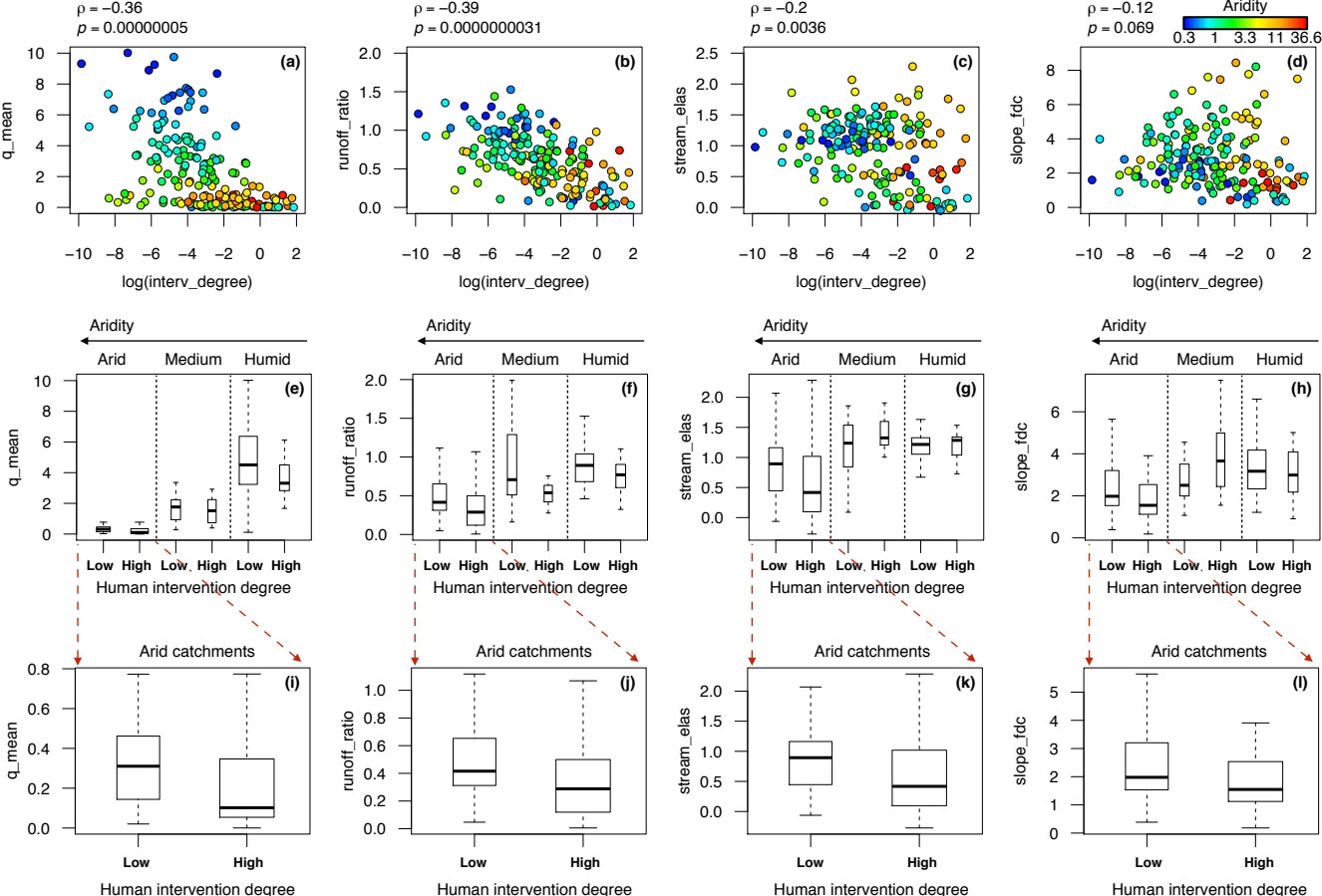

**Figure 14: Panels a-d present the relation between four hydrological signatures and the log-transformed human intervention degree ("interv_degree" from Table 3). The spearman rank correlation coefficients and their p-values at 95 % confidence are shown in each plot. The colour corresponds to the aridity index. Panels e-h show the boxplots (box widths are proportional to the number of catchments in each box) of the hydrological signatures for the catchments classified by their aridity (humid: aridity below 0.8, medium: aridity between 0.8 and 1.5, and arid: aridity above 1.5) and by their human intervention degree (low: interv_degree below 5 %, and high: interv_degree greater than 5 %). Panels i-l present a zoomed view of the arid catchments.**

**Tables**

**Table 1: Precipitation products**

| Name | Description | Spatial resolution | Temporal resolution | Period of record |
|---|---|---|---|---|
| precip$_{cr2met}$ | Obtained from de CR2METv1.3 dataset (DGA, 2017) | 0.05° lat-lon | daily | 1979-2016 |
| precip$_{tmpa}$ | Obtained from TMPA 3B42v7 dataset (Huffman et al., 2007, 2010) | 0.25° lat-lon | daily | 1998-2016 |
| precip$_{chirps}$ | Obtained from Climate Hazards Group InfraRed Precipitation with Station data (CHIRPS) version 2 dataset (Funk et al., 2015) | 0.05° lat-lon | daily | 1981-2016 |
| precip$_{mswep}$ | Obtained from the Multi-Source Weighted-Ensemble Precipitation (MSWEP) v1.1 dataset (Beck et al., 2017) | 0.25° lat-lon | daily | 1979-2016 |

5    **Table 2: Summary of attributes computed in CAMELS and CAMELS-CL.**

| Attribute class | CAMELS (A17) | CAMELS-CL |
|---|---|---|
| Location and topography | 9 attributes | 6 attributes adopted from A17<br>11 additional attributes |
| Geology | 7 attributes | 5 attributes adopted from A17 |
| Soils characteristics | 11 attributes | not computed |
| Land cover characteristics | 8 attributes | 3 attributes adopted from A17<br>13 additional attributes |
| Climatic indices | 11 attributes | 11 attributes adopted from A17<br>1 additional attribute |
| Hydrological signatures | 13 attributes | 13 attributes adopted from A17<br>1 additional attribute |
| Intervention degree | not computed | 6 attributes |
| **Total number of attributes** | **59** | **38 adopted from A17**<br>**32 introduced** |

**Table 3: Summary of catchment attributes. Climate indices and hydrological signatures were computed for the period 01/01/1990-31/12/2010. Index i refers to the precipitation product (i = 1, 2, 3, 4 for precip$_{cr2met}$, precip$_{tmpa}$, precip$_{chirps}$ and precip$_{mswep}$, respectively).**

| Attribute class | Attribute name | Description | Unit | Data source | Reference |
|---|---|---|---|---|---|
| Location and topography | gauge_id | catchment identifier (corresponds to the station code provided by DGA) | - | Gauges information collected from http://explorador.cr2.cl | Section 3.1 |
| | gauge_name | gauge name (based on DGA records) | - | | |
| | gauge_lat | gauge latitude (based on DGA records) | ° South | | |
| | gauge_lon | gauge longitude (based on DGA records) | ° West | | |
| | gauge_record_start | start date of streamflow records | - | | |
| | gauge_record_end | end date of streamflow records | - | | |
| | gauge_n_obs | number of days with valid streamflow records | day | | |
| | area | catchment area | km$^2$ | ASTER GDEM 30 m raster data (Tachikawa et al., 2011) | |
| | elev_gauge | gauge elevation (catchment outlet) obtained from the 30 m ASTER GDEM elevation data and the location provided by DGA | m a.s.l. | | |
| | elev_mean | catchment mean elevation | m a.s.l. | | |
| | elev_med | catchment median elevation | m a.s.l. | | |
| | elev_max | catchment maximum elevation | m a.s.l. | | |
| | elev_min | catchment minimum elevation | m a.s.l. | | |
| | slope_mean | catchment mean slope | m km$^{-1}$ | | |
| | nested_inner | number of inner catchments contained within gauge_id catchment (the gauge_id for the inner catchments can be obtained from the hierarchy matrix described in Sect. 3.1.1) | - | - | |
| | nested_outer | number of catchments containing gauge_id catchment (the gauge_id for the outer catchments can be obtained from the hierarchy matrix described in Sect. 3.1.1) | - | | |
| | location_type | classification in "coastal (or low elevation)", "foothill" and "altiplano" catchments, based on gauge elevations (gauge_elev) below 50 m a.s.l., between 1000 and 1200 m a.s.l., and above 3,500 m a.s.l., respectively. | - | - | Section 3.2 |
| Geological characteristics | geol_class_1st | most common geologic class in the catchment | - | Global Lithological Map database (GLiM) (Hartmann and Moosdorf, 2012) | Table 6 in A17 |
| | geol_class_1st_frac | fraction of the catchment area associated with its most common geologic class | - | | |
| | geol_class_2nd | second most common geologic class in the catchment | - | | |
| | geol_class_2nd_frac | fraction of the catchment area associated with its second most common geologic class | - | | |
| | carb_rocks_frac | fraction of the catchment area characterised as "carbonate sedimentary rocks" | - | | |
| Land cover characteristics | crop_frac | percentage of the catchment covered by croplands, level 1. Includes five types of level 2 classes: rice fields; greenhouse farming; other croplands; orchards; and bare croplands | % | 30 m resolution land cover map provided by Zhao et al. (2016) | Sections 3.1.3 and 3.2.3 |
| | nf_frac | percentage of the catchment covered by forest (level 1) classified as natural broadleaf (level 2) or natural conifer (level 2). | % | | |
| | fp_frac | percentage of the catchment covered by forest (level 1) classified as broadleaf plantations (level 2) or conifer plantations (level 2). | % | | |
| | grass_frac | percentage of the catchment covered by grasslands, level 1. Includes three types of level 2 classes: pastures; other grasslands; and withered grasslands. | % | | |
| | shrub_frac | percentage of the catchment covered by shrublands, level 1. Includes five types of level 2 classes: shrublands; shrubs and sparse trees mosaic; succulents; shrub plantations; and withered shrublands. | % | | |
| | wet_frac | percentage of the catchment covered by wetlands and water bodies (level 1). Includes six types of level 2 classes: marshlands; mudflats; other wetlands; lakes; reservoirs/ponds; rivers; and ocean. | % | | |
| | imp_frac | percentage of the catchment covered by impervious surfaces (level 1). Urbanised areas are usually contained in this class. | % | | |
| | barren_frac | percentage of the catchment covered by barren lands (level 1). Includes three types of level 2 classes: dry salt flats; sandy areas; and bare exposed rocks | % | | |
| | snow_frac | percentage of the catchment covered by snow and ice, level 1. Includes two types of level 2 classes: snow and ice. | % | | |

| | | | | |
|---|---|---|---|---|
| | fp_nf_index | forest plantation index: calculated as the ratio between fp_frac and the total forested area (fp_frac+nf_frac). | - | |
| | forest_frac | fraction of the catchment covered by forests, including native forest and forest plantation (fp_frac+nf_frac). | % | |
| | dom_land_cover | dominant land cover class | - | |
| | dom_land_cover_frac | fraction of the basin associated with dominant land cover class | % | |
| | land_cover_missing | percentage of the basin not covered by the land cover map | % | |
| | glaciers_area | glacierised area within the catchment | km$^2$ | Randolph Glacier Inventory v. 6.0 (RGI Consortium, 2017) | Sections 3.1.4 and 3.2.3 |
| | glaciers_frac | percentage of the catchment covered by glaciers. | % | | |
| Climatic indices (computed for 1 April 1990 to 31 March 2010) | p_mean_i | mean daily precipitation of product $i$ | mm day$^{-1}$ | Precipitation, temperature and potential evapotranspiration products introduced in Sect. 3.1.6, 3.1.7 and 3.1.8, respectively. | Table 2 in A17 |
| | p_mean_spread | coefficient of variation of basin-averaged mean annual precipitation from different products (standard deviation of p_mean_i from the four precipitation products, normalised by multi-product mean) | - | | |
| | pet_mean | mean daily PET of pet$_{har}$ product | mm day$^{-1}$ | | |
| | aridity_i | aridity, calculated as the ratio of mean daily PET (pet_mean) to mean daily precipitation (p_mean_i) | - | | |
| | p_seasonality_i | seasonality and timing of precipitation (product $i$) estimated using sine curves to represent the annual temperature and precipitation cycles; positive (negative) values indicate that precipitation peaks in summer (winter); values close to 0 indicate uniform precipitation throughout the year | - | | |
| | frac_snow_i | fraction of precipitation (product $i$) falling as snow (i.e., on days colder than 0 ◦C) | - | | |
| | high_prec_freq_i | frequency of high precipitation days ($\geq$ 5 times mean daily precipitation) for product $i$ | days yr$^{-1}$ | | |
| | high_prec_dur_i | average duration of high precipitation events (number of consecutive days $\geq$ 5 times mean daily precipitation), for product $i$ | days | | |
| | high_prec_timing_i | season during which most high precipitation days ($\geq$ 5 times mean daily precipitation) occur | season | | |
| | low_prec_freq_i | frequency of dry days (< 1 mmday$^{-1}$), for product $i$ | days yr$^{-1}$ | | |
| | low_prec_dur_i | average duration of dry periods (number of consecutive days <1 mmday$^{-1}$), for product $i$ | days | | |
| | low_prec_timing_i | season during which most dry days (< 1 mmday$^{-1}$) occur, for product $i$ | season | | |
| Hydrological signatures (computed for 1 April 1990 to 31 March 2010) | q_mean | mean daily discharge | mm day$^{-1}$ | Streamflow records collected from http://explorador.cr2.cl | Table 3 in A17 |
| | runoff_ratio_i | runoff ratio (ratio of mean daily discharge to mean daily precipitation), for product $i$ | - | | |
| | stream_elas_i | streamflow precipitation elasticity (sensitivity of streamflow to changes in precipitation at the annual timescale, using the mean daily discharge as reference and precipitation product $i$) | - | | |
| | slope_fdc | slope of the flow duration curve, FDC (between the log- transformed 33rd and 66th streamflow percentiles) | - | | |
| | baseflow_index | baseflow index (ratio of mean daily baseflow to mean daily discharge, hydrograph separation performed using the Ladson et al. (2013) digital filter with α set to 0.925) | - | | |
| | hdf_mean | mean half-flow date (date on which the cumulative discharge since 1 April reaches half of the annual discharge) | day of the year | | |
| | Q5 | 5 % flow quantile (low flows) | mm day$^{-1}$ | | |
| | Q95 | 95 % flow quantile (high flows) | mm day$^{-1}$ | | |
| | high_q_freq | frequency of high-flow days (> 9 times the median daily flow) | days yr$^{-1}$ | | |
| | high_q_dur | average duration of high-flow events (number of consecutive days >9 times the median daily flow) | days | | |
| | low_q_freq | frequency of low-flow days (< 0.2 times the mean daily flow) | days yr$^{-1}$ | | |
| | low_q_dur | average duration of low-flow events (number of consecutive days <0.2 times the mean daily flow) | days | | |
| | zero_q_freq | percentage of days with Q=0 | % | | |

| | | | | | |
|---|---|---|---|---|---|
| | swe_ratio | ratio of peak of snow water equivalent to mean annual discharge | - | SWE product developed by Cortés and Margulis (2017) | |
| Intervention | sur_rights_n | total number of granted surface rights within the catchment | - | Water Atlas developed by the DGA (DGA, 2016a) | Sections 3.1.10 and 3.2.6 |
| | sur_rights_flow | annual flow calculated for consumptive permanent continuous surface water rights | $m^3\ s^{-1}$ | | |
| | interv_degree | intervention degree defined as the annual flow of surface water rights (consumptive permanent continuous), normalised by mean annual streamflow. | - | | |
| | gw_rights_n | total number of granted groundwater rights within the catchment | - | | |
| | gw_rights_flow | annual flow calculated for consumptive permanent continuous groundwater water rights | $m^3\ s^{-1}$ | | |
| | large_dam | 0 if there is no dam within the catchment, 1 if there is at least one dam classified as "large" | - | http://www.ide.cl/descarga/capas/item/embalses-2016.html | |

**Table 4: Evaluation metrics of PET gridded products (PET$_{har}$ and PET$_{mod}$). Pearson correlation coefficients (r) and the ratios between gridded PET and observation-based PET (INIA$_{har}$ and INIA$_{ETO}$) were spatially averaged within the macro-zones.**

| Macro-zone | PET$_{har}$ compared with INIA$_{har}$ | | PET$_{mod}$ compared with INIA$_{ET0}$ | |
|---|---|---|---|---|
| | r | ratio | r | ratio |
| Far North | 0.76 | 0.96 | 0.19 | 1.66 |
| Near North | 0.92 | 0.92 | 0.91 | 1.68 |
| Central Zone | 0.95 | 1.00 | 0.96 | 1.79 |
| Southern Zone | 0.97 | 1.07 | 0.96 | 1.58 |
| Austral Zone | 0.97 | 1.08 | 0.95 | 1.14 |
| Southern Patagonia | 0.98 | 0.93 | 0.96 | 1.06 |