# Peer review of "The CAMELS-CL dataset: catchment attributes and meteorology for large sample studies – Chile dataset"

_Hydrology and Earth System Sciences, 2018_

## Referee Comment (RC1) · Anonymous Referee #1 · 8 Apr 2018

This paper presents the first large-sample catchment dataset for Chile with many data sources compiled to provide an easy to use, uniform dataset. The data sources and derived statistics are clearly described in sections 3 and 4 and the authors generally discuss uncertainty and caveats to each piece of the dataset. A valuable addition is the compilation of diversion data for the catchments. Overall, the paper is easy to read and the figures are high quality. However, it reads more like a data paper with relevant descriptions of the components of a dataset. Section 5 does provide some nice analysis, but more is needed before publication. Additionally, another careful pass through the paper for typos and minor grammar issues is needed.

Specific comments:

1) This paper lacks substantial analysis. The new dataset is very valuable, but as the manuscript stands now, it is very close to a data paper. Additional analysis is needed.

One opportunity is uncertainty. The authors note many times that the input climate data, streamflow, and basin boundary data are uncertain. The authors could (should?) provide uncertainty bounds on the derived indices (e.g. runoff ratio) using the various climate indices and possibly any other known sources of uncertainty. An interesting aspect that could be explored is quality control of the various climate data using metrics like runoff ratio to identify suspect datasets (e.g. long-term runoff ratio > 1 in areas of little glacial melt or groundwater).

This can then be parlayed into analysis on how uncertainty may impact the significance of any spatial trends in the catchment hydrologic signatures or climate indices. This does not need to be exhaustive, but a nice example of how to use the uncertainty data provided would be useful to the community.

2) The results of section 5 seem very logical, yet I don't see any citations in the paper discussing basin response with human intervention. There must be some previous work that can be cited?

3) Additionally, this paper appears to only focuses on diversions, not urban fraction. Catchment response changes with urban fraction as well, which is obviously another component of human intervention in watersheds. If this is negligible in these catchments, the authors should note it.

4) The concluding remarks section is very repetitive. Key contributions from this paper are restated several times in different ways across the section and should be condensed.

---

## Referee Comment (RC2) · Anonymous Referee #2 · 23 Apr 2018

The paper describes a new dataset of catchment attributes for Chile, as well as a short analysis using this dataset. The dataset itself should be very useful to those studying hydrology in the area. Similar datasets in other regions have proven to be enormously useful and widely used. The paper provides a good description of the attributes included in the dataset. In addition, the analysis provides a good overview of hydrologic and land use conditions in Chile, with a few observations about correlation between different variables.

A few more specific comments:

- In the description of water rights, many different types of water use are mentioned.

[Figure]

Are there types or amounts of water use that do not require a permitted water right? (For example, below a threshold amount?)

- There is a missing band of elevation between 1000 m.a.s.l. and 3500 m.a.s.l, if I understand the categorizations described in 4.1 correctly. Why is this?

- Section 5 and 6: be careful to be precise in the wording so as not to confuse correlation and causation. (See sentences page 17, line 16; page 28, line 29)

- Throughout the paper, there are a number of small grammatical issues that could be improved by a close reading.

I recommend publication of this paper as a means of making this dataset available and as a demonstration of its utility, provided this type of paper is within the scope of HESS. Note that the analysis portion of this paper is not extensive.

---

## Author Comment (AC1) · 17 May 2018

**"The CAMELS-CL dataset: catchment attributes and meteorology for large sample studies – Chile dataset" by Alvarez-Garreton et al.**

Response to Anonymous Referees #1 and #2

We thank anonymous Referees #1 and #2 for their constructive comments on our paper. Before we address them, we would like to mention that since our manuscript was submitted, we have updated some datasets and added a new one. Firstly, 59 catchment boundaries (and their corresponding hydro-meteorological time series and catchment attributes) were updated based on more accurate locations of streamflow gauges. This new location information was provided by technicians from the Chilean Water Directorate (DGA). Secondly, we incorporated an updated water rights dataset developed for the National Water Balance project (DGA, 2017), where several water rights with missing coordinates in the former public database used in CAMELS-CL were located by using Google Earth Imagery. Finally, we added a public dataset of national dams (http://www.ide.cl/descarga/capas/item/embalses-2016.html) to identify the presence of dams within catchments. This was quantified as a binary attribute (0 if there is no dam within the catchment, and 1 if there is at least one dam) that can be used to select near-natural catchments. We think that these additions significantly improve the quality of CAMELS-CL dataset. The updated time series and catchment attributes are freely available online: http://www.cr2.cl/download/camels-cl/.

Below we provide responses to each Referee comment. For clarity, Referee comments are given in blue text, our responses are given in plain text and the proposed paper modifications in italics.

Referee #1:

This paper presents the first large-sample catchment dataset for Chile with many data sources compiled to provide an easy to use, uniform dataset. The data sources and derived statistics are clearly described in sections 3 and 4 and the authors generally discuss uncertainty and caveats to each piece of the dataset. A valuable addition is the compilation of diversion data for the catchments. Overall, the paper is easy to read and the figures are high quality.

We thank the reviewer for her/his positive comments.

However, it reads more like a data paper with relevant descriptions of the components of a dataset. Section 5 does provide some nice analysis, but more is needed before publication.

We agree that an important part of the paper is related to the description and discussion of the input datasets used to generate CAMELS-CL. Nevertheless, in our view the two scientific questions addressed by the paper are in line with HESS main scopes: 1) "*the study of the spatial and temporal characteristics of the global water resources (...)*" and 2) "*the study of interactions with human activity of all the processes, budgets, fluxes (..)*" (https://www.hydrology-and-earth-system-sciences.net/about/aims_and_scope.html). The first scope frames the scientific question addressed in Section 4, where we analyse the spatial distribution of catchment attributes, providing insights about catchment hydrologic response and dominant runoff processes, and their relation with catchment characteristics and climatic indices. The second scope frames the analysis on the impact of human intervention on catchment hydrologic response (addressed in Section 5).

In response to the reviewer's recommendation, we propose to add a new section to the manuscript where we analyse one of the main sources of uncertainty of the dataset (*New Section 6: Precipitation uncertainty analysis*). The proposed section is presented at the end of this document.

Additionally, another careful pass through the paper for typos and minor grammar issues is needed.

The complete manuscript has been proof read to correct typos and grammar issues.

Specific comments:

1) This paper lacks substantial analysis. The new dataset is very valuable, but as the manuscript stands now, it is very close to a data paper. Additional analysis is needed. One opportunity is uncertainty. The authors note many times that the input climate data, streamflow, and basin boundary data are uncertain. The authors could (should?) provide uncertainty bounds on the derived indices (e.g. runoff ratio) using the various climate indices and possibly any other known sources of uncertainty.

We have considered the reviewer's recommendation and performed an analysis of forcing uncertainty coming from the precipitation products in CAMELS-CL. This analysis is proposed as a new section (*New Section 6: Precipitation uncertainty analysis*), where we quantify the spread from all precipitation products and discuss potential relationships with

physical catchment characteristics and catchment hydrologic response. This forcing spread will be incorporated as a new catchment attribute (a new climatic index called "p_mean_spread").

An interesting aspect that could be explored is quality control of the various climate data using metrics like runoff ratio to identify suspect datasets (e.g. long-term runoff ratio > 1 in areas of little glacial melt or groundwater). This can then be parlayed into analysis on how uncertainty may impact the significance of any spatial trends in the catchment hydrologic signatures or climate indices. This does not need to be exhaustive, but a nice example of how to use the uncertainty data provided would be useful to the community.

We thank the reviewer for this excellent idea. We computed runoff ratios for near-natural catchments to evaluate the different forcing products. Based on this, we provide insights on which regions and catchment characteristics are more prone to precipitation errors (*New Section 6: Precipitation uncertainty analysis*).

2) The results of section 5 seem very logical, yet I don't see any citations in the paper discussing basin response with human intervention. There must be some previous work that can be cited?

The results are indeed logical and intuitive. However, they are not commonly found in large sample studies (over 100 catchments) using water extractions to quantify the degree of human intervention. We propose to add the following references to Section 5, where different types of human intervention are related to catchment response:

*Poff et al. (2006) examined the effects of land use on hydrological regimes (e.g., peak and low flows, runoff variability) in 158 basins across the conterminous United States, finding region-dependent changes in specific metrics. Ochoa-Tocachi et al. (2016) analysed the impacts of land use on the hydrology of 25 Andean catchments, finding that anthropogenic influence translates into increased streamflow variability and decreased catchment regulation capacity and water yield. More recently, Tijdeman et al. (2018) examined the effects of human intervention on streamflow drought characteristics across 187 catchments in England and Wales, concluding that most human-influenced catchments did not have flow drought characteristics different from those expected for near-natural conditions.*

*References:*
*Poff, N. L., Bledsoe, B. P., & Cuhaciyan, C. O. (2006). Hydrologic variation with land use across the contiguous United States: geomorphic and ecological consequences for stream ecosystems. Geomorphology, 79(3-4), 264-285.*
*Ochoa-Tocachi, B. F., Buytaert, W., De Bièvre, B., Célleri, R., Crespo, P., Villacís, M., ... & Gil-Ríos, J. (2016). Impacts of land use on the hydrological response of tropical Andean catchments. Hydrological processes, 30(22), 4074-4089.*
*Tijdeman, E., Hannaford, J., & Stahl, K. (2018). Human influences on streamflow drought characteristics in England and Wales. Hydrology and Earth System Sciences, 22(2), 1051.*

3) Additionally, this paper appears to only focuses on diversions, not urban fraction. Catchment response changes with urban fraction as well, which is obviously another component of human intervention in watersheds. If this is negligible in these catchments, the authors should note it.

The reviewer is right. Urban fraction is another important factor modulating catchment response, and it can be obtained from land cover attributes, since urban areas are usually classified as impervious surfaces. In the submitted manuscript, the imp_frac attribute (Table 3) included two classes: impervious surfaces and barren lands. To enable the quantification of urbanised areas within a catchment, we propose to compute these classes as two different land cover attributes: imp_frac and barren_frac. By doing this, the urban fraction of the catchments (assumed to be equal than the impervious area, imp_frac) varies between 0% and 7% for most catchments (only one catchment had imp_frac = 25%). This is considered negligible and it will be stated in a revised Section 4.3 (Land cover attributes). It should be noted, however, that there is uncertainty in this estimation. Zhao et al., (2016) – whose land cover map was used to produce CAMELS-CL – highlighted that the impervious surface class is the worst classified class, since urban areas have mixed pixels of vegetation and paved surfaces. This discussion will be also added to revised Section 4.3.

4) The concluding remarks section is very repetitive. Key contributions from this paper are restated several times in different ways across the section and should be condensed.

The conclusion section was condensed, deleting repetitions.

Referee #2:

The paper describes a new dataset of catchment attributes for Chile, as well as a short analysis using this dataset. The dataset itself should be very useful to those studying hydrology in the area. Similar datasets in other regions have proven to be enormously useful and widely used. The paper provides a good description of the attributes included in the dataset. In addition, the analysis provides a good overview of hydrologic and land use conditions in Chile, with a few observations about correlation between different variables.

We thank the reviewer for her/his positive comments.

A few more specific comments:
- In the description of water rights, many different types of water use are mentioned. Are there types or amounts of water use that do not require a permitted water right? (For example, below a threshold amount?)

Strictly speaking, the Chilean Water Code (Congreso De La República De Chile, 1981) states that any water use should be associated with a water right allocated by the Chilean Water Directorate (DGA), even for low volumes. However, there are many extractions that are not informed to DGA and are not detected in subsequent inspections (due to lack of human resources devoted to carry out field inspections), which is the main challenge of quantifying human intervention within a catchment. This will be clarified in Section 3.11 (Water rights).

Reference:
Congreso De La República De Chile, 1981: Decreto Fuerza de Ley 1122. Fija Texto del Código de Aguas. 1–68.

- There is a missing band of elevation between 1000 m.a.s.l. and 3500 m.a.s.l, if I understand the categorizations described in 4.1 correctly. Why is this?

Thanks for pointing this. There was an error in the elevation bands description in Section 4.1. The text will be now consistent with Table 3:

*Section 4.1, page 12, line 13: We proposed the attribute location_type (see Table 3 and Fig. 7f) with three categories: coastal (or low elevation), foothills and altiplano catchments, defined by gauge elevations lower than 50 m a.s.l., between*  1000 and 1200 *m a.s.l., and above 3,500 m a.s.l., respectively.*

- Section 5 and 6: be careful to be precise in the wording so as not to confuse correlation and causation. (See sentences page 17, line 16; page 18, line 29)

We agree. Our analyses demonstrate correlation, but do not permit to establish causation. The wording in Sections 5 and 6 will be carefully revised to reflect this. In particular, the sentences mentioned by the Referee will be changed as follows:

*Section 5, page 17, line 16: Larger human intervention*  is correlated with *decreased annual flows and runoff ratios, especially in drier catchments.*

*Section 6, page 18, line 29: The analysis on the impacts of human activities on catchment behaviour leads to several important insights. We showed that larger human intervention*  is correlated with *decreased annual flows, runoff ratios, decreased elasticity of runoff with respect to precipitation, and decreased flashiness of runoff, especially in drier catchments.*

- Throughout the paper, there are a number of small grammatical issues that could be improved by a close reading.

The complete manuscript has been proof read to correct typos and grammar issues.

I recommend publication of this paper as a means of making this dataset available and as a demonstration of its utility, provided this type of paper is within the scope of HESS. Note that the analysis portion of this paper is not extensive.

We are very pleased that the Referee appreciates the contributions of this study. We truly hope that the analyses of the paper about dominant spatial patterns of physical, climatic and hydrological attributes, impacts of human intervention on catchment hydrologic response, and the new analyses presented here (*New Section 6: Precipitation uncertainty analysis*), contain the scientific relevance needed for publication in HESS.

Sincerely,

Camila Alvarez Garreton

*New Section 6: Precipitation uncertainty analysis*

*We provide a first-order analysis of inter-product differences across the domain. To this end, we define and compute a precipitation spread attribute (p_mean_spread) for each catchment as the coefficient of variation of the mean annual precipitation from different products (standard deviation of mean annual precipitation values from the four precipitation products, normalised by their mean). To allow the comparison of the four products, we use data for the common record period 1998-2014 (Table 2). The spread attribute represents differences among the various precipitation estimates, which can be associated with precipitation uncertainty. The underlying assumption is that similar values from different sources indicate regions with higher confidence in precipitation estimates.*

*Fig. x1 displays catchment-scale mean precipitation and the precipitation spread index for three main regions: North (northern than 34°S), which includes the Far North and Near North macrozones; Central-South (between 34°S and 44°S); and Austral-Patagonia (southern than 44°S). Mean precipitation (p_mean) estimates have a larger spread in the North (top and middle rows in the boxplots in Fig. 1x. Note the different scale use for p_mean in the North), mainly due to the low mean annual precipitation values typical in these (hyper-) arid regions. Although the absolute difference between products is low, the normalisation by a low mean value increases their dispersion. By contrast, considerably larger precipitation amounts and lower spread values are obtained in Central-South and Austral-Patagonia. However, it should be noted that the ensemble spread of precipitation estimates does not necessarily quantify the accuracy associated to each product, which is typically assessed using ground observations as the "truth" (e.g., Zambrano-Bigiarini et al. 2017). Such analysis is beyond the scope of this article, because the assessment of different precipitation products at the basin scale is typically carried out by forcing (one or more) hydrological model(s) with the different precipitation datasets over the selected study area (e.g., Bisselink et al. 2016; Thiemig et al. 2013; Su et al. 2008).*

*As an alternative to the model-based approach, we examine the consistency of catchment precipitation estimates based on the water balance at the basin outlet, which is quantified by the long-term runoff ratio in 119 near-natural catchments (bottom row in the boxplots in Fig. x1). These catchments share the following characteristics: interv_degree lower than 0.1 (less than 10% of the annual streamflow allocated to surface rights), large_dam equal to zero (no presence of large dams within the catchment), glacier_frac lower than 5% (negligible glacier contribution), imp_frac lower than 5% (negligible urban areas), and copr_frac lower than 20% (negligible irrigation effects). Although there is large dispersion of runoff ratios in the North (consistent with large p_mean_spread values), there are no inconsistent runoff ratios within this domain. By contrast, there are catchments with runoff ratios larger than 1 in Central-South and Austral-Patagonia, indicating that more water is leaving the catchment than the total amount entering as precipitation. Assuming that streamflow data and catchment area are reliable, such cases indicate precipitation underestimation.*

[Figure]

*Figure x1: (Left) Precipitation spread (p_mean_spread in top row), mean annual precipitation (middle row), and runoff ratio (bottom row), for the different precipitation products. (Right) Distribution of p_mean_spread in three main regions: North (northern than 32°S); Central-South; and Austral-Patagonia.*

In the Central-South region, the MSWEP dataset provides the smallest percentage of catchments (9%) with runoff ratio > 1, whereas in the Austral-Patagonia domain, the product CR2MET provides the smallest percentage (15%). In both domains, the TMPA dataset provides the largest percentage of catchments with runoff ratio > 1 (55% in Central-South and 66% in Austral-Patagonia). The precipitation underestimation in TMPA product was also found by Zambrano-Bigiarini et al. (2017).

A natural question that arises is: what are the main catchment characteristics associated with inter-forcing differences? Figure x2 shows the relationship between runoff ratios and catchment outlet latitude (first column), maximum catchment elevation (second column), and mean catchment slope (third column). These three attributes are the strongest predictors among all topographic, geologic and land cover attributes (Table 3) used in a generalised linear model (not shown). Because runoff mechanisms are highly non-linear, we use the Spearman rank correlation coefficient to explore the relationship between runoff ratios and these attributes. The results in Figure x2 show that runoff ratios greater than one (pink markers) are associated with catchments southern than 34°S (consistent with Fig. x1), showing a significant (p-value lower than 0.05) negative correlation with latitude (first column in Fig. x2).

Since only runoff ratios larger than 1 are indicative of precipitation under-estimation (lower than one informs that the water balance is not violated, but does not provide further information about precipitation quality), we pay close attention on Central-South and Austral-Patagonia regions. To this aim, the correlations with elevation (second column in Fig. 2x) and slope (third column) are computed disregarding Northern catchments (i.e., yellow markers). These plots show significant positive correlation coefficients between runoff ratios and maximum catchment elevation and slope, with runoff ratio values above one for all precipitation products (one product per row). Further, one can note that Northern catchments (yellow symbols) behave very differently in terms of hydroclimatology compared to southern ones (blank and pink markers). Indeed, if these catchments were included in the analysis, the correlation would not be positive nor statistically significant, which supports the idea of clustering the catchments for the analysis.

[Figure]

Figure x2: Correlation between runoff ratios calculated for the different precipitation products and outlet latitude (first column), maximum elevation of the catchment (second column), mean slope of the catchment (third column). The Spearman correlation coefficient (ρ), p-value and fitted regression curves are presented in each plot. Pink markers represent runoff ratios above one. Yellow markers represent catchments with gauge latitude northern than 34°S.

In summary, the above results provide two main insights. Firstly, there is large precipitation uncertainty in the Far North and Near North regions due to the relatively rare occurrence of precipitation events. These meteorological characteristics pose methodological challenges for detecting events and estimating their intensities. Although the implications of precipitation uncertainty on streamflow modelling are not straightforward to assess in this sub-domain, some insights can be gained from previously presented results. For example, we have showed that surface runoff is not very sensitive to variations in precipitation (see low values of runoff elasticity to precipitation in Fig. 11f), which may suggest a weak

transferring of precipitation errors within a model. On the other hand, groundwater has the largest contribution to streamflow in this domain (largest baseflow indices in Fig. 11e and sedimentary rocks as the most common geologic class illustrated in Fig. 8a), which suggests that an important source of uncertainty would be related to the representation of groundwater mechanisms in a model. Furthermore, aquifer boundaries may be different from the topography-based catchment delineation performed here, thus a sound identification of the existing aquifers should be carried out to ensure a good representation of the surface-groundwater interaction (e.g., Sar et al. 2015; Arkoprovo et al. 2012; Ivkovic et al. 2009).

The second insight from this uncertainty analysis is that all precipitation products tend to underestimate precipitation in catchments located south of 32°S, featuring high elevations and steep slopes (the general case for headwater catchments in the Andes). Such limitation is well known and it may be attributed to several factors, including the complex topography of mountain catchments and the corresponding orographic effects that are not fully accounted for at the satellite and reanalysis grid resolution. Another reason is the lack of ground-based data in headwater catchments (90% of the 500 rain gauges located south of 32°S are placed below 1,000 m a.s.l.), a critical issue considering that elevations in Chile goes up to 7,000 m a.s.l. and most of the water feeding the whole system occurs there. This poses challenges for streamflow modelling, water management and allocation, since it restricts the full understanding of hydrological processes. To alleviate these limitations, more monitoring stations are needed at high elevations (above 2,000 or 1,500 m a.s.l. at least) to improve remotely-sensed and model-based precipitation estimates in complex terrains.

New References:

Arkoprovo, B., J. Adarsa, and S. S. Prakash, 2012: Delineation of Groundwater Potential Zones using Satellite Remote Sensing and Geographic Information System Techniques : A Case study from Ganjam district , Orissa , India. Res. J. Recent Sci., **1**, 59–66.

Bisselink, B., M. Zambrano-Bigiarini, P. Burek, and A. de Roo, 2016: Assessing the role of uncertain precipitation estimates on the robustness of hydrological model parameters under highly variable climate conditions. J. Hydrol. Reg. Stud., **8**, 112–129, doi:10.1016/j.ejrh.2016.09.003.

DGA, 2017: Actualizacion del Balance Hídrico Nacional.

Ivkovic, K. M., R. A. Letcher, and B. F. W. Croke, 2009: Use of a simple surface-groundwater interaction model to inform water management. Australian Journal of Earth Sciences, Vol. 56 of, 71–80.

Sar, N., A. Khan, S. Chatterjee, and A. Das, 2015: Hydrologic delineation of ground water potential zones using geospatial technique for Keleghai river basin, India. Model. Earth Syst. Environ., **1**, 25, doi:10.1007/s40808-015-0024-3. http://link.springer.com/10.1007/s40808-015-0024-3.

Su, F., Y. Hong, and D. P. Lettenmaier, 2008: Evaluation of TRMM Multisatellite Precipitation Analysis (TMPA) and Its Utility in Hydrologic Prediction in the La Plata Basin. J. Hydrometeorol., **9**, 622–640, doi:10.1175/2007JHM944.1. http://journals.ametsoc.org/doi/abs/10.1175/2007JHM944.1.

Thiemig, V., R. Rojas, M. Zambrano-Bigiarini, and A. De Roo, 2013: Hydrological evaluation of satellite-based rainfall estimates over the Volta and Baro-Akobo Basin. J. Hydrol., **499**, 324–338, doi:10.1016/j.jhydrol.2013.07.012.

Zambrano-Bigiarini, M., A. Nauditt, C. Birkel, K. Verbist, and L. Ribbe, 2017: Temporal and spatial evaluation of satellite-based rainfall estimates across the complex topographical and climatic gradients of Chile. Hydrol. Earth Syst. Sci., **21**, 1295–1320, doi:10.5194/hess-21-1295-2017.

---

## Author Response (AR1)

**"The CAMELS-CL dataset: catchment attributes and meteorology for large sample studies – Chile dataset" by Alvarez-Garreton et al.**

Response letter to Anonymous Referees #1 and #2

We thank the Editor, and anonymous Referees #1 and #2 for their constructive comments on our paper. By following their recommendations, we have significantly improved the scientific quality of this work.

In general, both Referees agreed in the great value of CAMELS-CL and its potential use in hydrological applications. They were consistent in qualifying the manuscript as well written, with clear descriptions of the datasets and with high-quality figures. Referee #1 and the Editor recommended to deepen the data analysis of the paper, which we took as the main objective to fulfil in this revision process. To accomplish this, we re-structured the manuscript and incorporated new analyses, making more explicit the double character of this study: a dataset presentation and scientific applications. The revised Sect. 3 now presents the CAMELS-CL dataset, including the description of the input datasets (Sect. 3.1) and the derived catchment attributes with the discussion of their spatial distributions (Sect. 3.2). We created a new Sect. 4, where we analyse the reliability of precipitation (4.1) and PET data (4.2) at the basin scale. Finally, in Section 5 we present a diagnostic of the impacts of human intervention on catchment behaviour.

The new analyses presented in Sect. 4 provide key insights about the quality of the different forcing products. In Sect. 4.1, we assessed one national and three global precipitation products. We showed large inter-product discrepancies in arid regions, which we related to the methodological challenges of detecting events and estimating their intensities in a domain with relatively rare occurrence of precipitation events. Further, we used Budyko curves to analyse the catchments water balance, and showed a systematic precipitation underestimation in headwater mountain catchments (high elevations and steep slopes) over humid regions, which we attributed to the complex topography of head-water catchments and the scarcity of ground stations at high elevations. These results were discussed and contrasted with the literature in Sect. 4.1, and the main conclusions were highlighted in the Abstract and Concluding remarks.

The two PET products (a daily PET calculated with Hargreaves formulae, using only air temperature data, and a MODIS 8-day accumulated product) were assessed in Sect. 4.2 by using an independent set of PET point values calculated from meteorological observations. We found good correlation coefficients ($r > 0.91$) for both products in humid regions and lower correlation ($r < 0.76$) in hyper-arid regions. The MODIS product showed a systematic overestimation across the domain, which was explained by the formulation differences with ground-based estimates. PET assessment results were discussed and contrasted with the literature in Sect. 4.2, and the main conclusions were highlighted in the Abstract and Concluding remarks (see revised manuscript).

Finally, in Sect. 5 we showed that anthropic intervention correlates with lower than normal annual flows, runoff ratios, elasticity of runoff with respect to precipitation, and flashiness of runoff, especially in arid catchments.

By incorporating these new elements, the contributions of the revised manuscript are: (i) to provide a unique dataset that can be used to advance our understanding of hydrological systems by learning from diversity, (ii) to discuss the dominant spatial patterns of physical, climatic and hydrological attributes within the domain, (iii) to evaluate one national and three global precipitation datasets based on the observed water balance, (iv) to assess the PET products based on an independent set of PET point values calculated from meteorological records, and (v) to examine the interplay between human intervention and changes in observed catchment response.

In addition, we would like to mention that since our manuscript was submitted, we have updated some datasets and added a new one. Firstly, 59 catchment boundaries (and their corresponding hydro-meteorological time series and catchment attributes) were updated based on more accurate locations of streamflow gauges. This new location information was provided by technicians from the Chilean Water Directorate (DGA). Secondly, we incorporated an updated water rights dataset developed for the National Water Balance project (DGA, 2017), where several water rights with missing coordinates in the former public

database used in CAMELS-CL were located by using Google Earth Imagery (Sect. 3.1.10). Finally, we added a public dataset of national dams (http://www.ide.cl/descarga/capas/item/embalses-2016.html) to identify the presence of dams within catchments. This was quantified as a binary attribute (0 if there is no dam within the catchment, and 1 if there is at least one dam) that can be used to select near-natural catchments (Sect. 3.2.6). We think that these additions have significantly improved the quality of CAMELS-CL dataset.

Below we provide responses to each Referee comment. We also provide a revised and marked-up manuscript, where all the changes and additions are highlighted.

Referee #1:

This paper presents the first large-sample catchment dataset for Chile with many data sources compiled to provide an easy to use, uniform dataset. The data sources and derived statistics are clearly described in sections 3 and 4 and the authors generally discuss uncertainty and caveats to each piece of the dataset. A valuable addition is the compilation of diversion data for the catchments. Overall, the paper is easy to read and the figures are high quality.

We thank the reviewer for her/his positive comments.

However, it reads more like a data paper with relevant descriptions of the components of a dataset. Section 5 does provide some nice analysis, but more is needed before publication.

Following the Referee recommendation, the revised manuscript incorporates an entire new section (Sect. 4: Uncertainty in precipitation and PET), where we analysed precipitation (Sect. 4.1) and PET (Sect. 4.2). This complements the analysis of the impacts of human intervention on catchment behaviour (Sect. 5).

The analyses of the revised Sect. 3.2 (Spatial distribution of derived catchment attributes), Sect. 4 and Sect. 5, are in line with HESS main scopes: 1) "*the study of the spatial and temporal characteristics of the global water resources (...)*" and 2) "*the study of interactions with human activity of all the processes, budgets, fluxes (..)*" (https://www.hydrology-and-earth-system-sciences.net/about/aims_and_scope.html).

To avoid extending the manuscript given the new analyses and new figures, we dismissed Fig. 1 of the former manuscript ("Figure 1: World map with topography at 1 km resolution (USGS, 1996). The Chilean boundary is highlighted in black"), which was initially designed to provide context on the singular topography that Chile represents compared to the rest of the world.

Additionally, another careful pass through the paper for typos and minor grammar issues is needed.

The complete manuscript has been proof read to correct typos and grammar issues.

Specific comments:

1) This paper lacks substantial analysis. The new dataset is very valuable, but as the manuscript stands now, it is very close to a data paper. Additional analysis is needed. One opportunity is uncertainty. The authors note many times that the input climate data, streamflow, and basin boundary data are uncertain. The authors could (should?) provide uncertainty bounds on the derived indices (e.g. runoff ratio) using the various climate indices and possibly any other known sources of uncertainty.

We have considered the reviewer's recommendation and performed a full analysis of forcing uncertainty (please see Sect. 4 in the revised manuscript). In particular, in Sect. 4.1 – following the reviewer's suggestion – we quantified the spread from all precipitation products as a measure of confidence bounds. We discussed the spatial distribution of this spread attribute and its relation with climatic conditions, providing valuable insights about the quality and limitations of the precipitation products.

An interesting aspect that could be explored is quality control of the various climate data using metrics like runoff ratio to identify suspect datasets (e.g. long-term runoff ratio > 1 in areas of little glacial melt or groundwater). This can then be parlayed into analysis on how uncertainty may impact the significance of any spatial trends in the catchment hydrologic signatures or climate indices. This does not need to be exhaustive, but a nice example of how to use the uncertainty data provided would be useful to the community.

We thank the reviewer for this idea. In Sect. 4.1, we analysed the runoff ratios for near-natural catchments to evaluate the different forcing products. With this, and by using Budyko curves, we identified regions where the different products were more prone to precipitation errors. We related these errors with catchment characteristics, climatic conditions, and the scarcity of ground observations.

2) The results of section 5 seem very logical, yet I don't see any citations in the paper discussing basin response with human intervention. There must be some previous work that can be cited?

The results are indeed logical and intuitive. Unfortunately, they are not commonly found in large sample studies (over 100 catchments) using water extractions to quantify the degree of human intervention. We added references to Sect. 5, where other types of human intervention are related to catchment response.

3) Additionally, this paper appears to only focuses on diversions, not urban fraction. Catchment response changes with urban fraction as well, which is obviously another component of human intervention in watersheds. If this is negligible in these catchments, the authors should note it.

We agree, urban fraction is another important factor modulating catchment response. This information can be obtained from land cover attributes, since urban areas are usually classified as impervious surfaces. In the submitted manuscript, the imp_frac attribute (Table 3) included two classes: impervious surfaces and barren lands. To enable the quantification of urbanised areas within a catchment, in the revised manuscript we computed these classes as two different land cover attributes: imp_frac and barren_frac. By doing this, the urban fraction of the catchments (assumed to be equal than the impervious area, imp_frac) varied between 0% and 7% for most catchments (only one catchment had imp_frac = 25%). This is considered negligible and it was stated in the revised Section 3.2.6 (Intervention). It should be noted, however, that there is uncertainty in this estimation. Zhao et al., (2016) – whose land cover map was used to produce CAMELS-CL – highlighted that the impervious surface class is the worst classified class, since urban areas have mixed pixels of vegetation and paved surfaces. This discussion was added to revised Section 3.2.6.

4) The concluding remarks section is very repetitive. Key contributions from this paper are restated several times in different ways across the section and should be condensed.

The conclusions were condensed, deleting repetitions.

Referee #2:

The paper describes a new dataset of catchment attributes for Chile, as well as a short analysis using this dataset. The dataset itself should be very useful to those studying hydrology in the area. Similar datasets in other regions have proven to be enormously useful and widely used. The paper provides a good description of the attributes included in the dataset. In addition, the analysis provides a good overview of hydrologic and land use conditions in Chile, with a few observations about correlation between different variables.

We thank the reviewer for her/his positive comments.

A few more specific comments:

- In the description of water rights, many different types of water use are mentioned. Are there types or amounts of water use that do not require a permitted water right? (For example, below a threshold amount?)

Strictly speaking, the Chilean Water Code (Congreso De La República De Chile, 1981) states that any water use should be associated with a water right allocated by the Chilean Water Directorate (DGA), even for low volumes. However, there are many extractions that are not informed to DGA and are not detected in subsequent inspections (due to lack of human resources devoted to carry out field inspections), which is the main challenge of quantifying human intervention within a catchment. This was clarified in Section 3.2.6 (Human intervention).

Reference: Congreso De La República De Chile, 1981: Decreto Fuerza de Ley 1122. Fija Texto del Código de Aguas. 1–68.

- There is a missing band of elevation between 1000 m.a.s.l. and 3500 m.a.s.l, if I understand the categorizations described in 4.1 correctly. Why is this?

Thanks for pointing this. There was an error in the elevation bands description in Section 3.2.1. The text was corrected.

- Section 5 and 6: be careful to be precise in the wording so as not to confuse correlation and causation. (See sentences page 17, line 16; page 18, line 29)

We agree. Our analyses show correlation, but do not permit to establish causation. The wording in Sections 5 and 6 were carefully revised to reflect this.

- Throughout the paper, there are a number of small grammatical issues that could be improved by a close reading.

The complete manuscript has been proof read to correct typos and grammar issues.

I recommend publication of this paper as a means of making this dataset available and as a demonstration of its utility, provided this type of paper is within the scope of HESS. Note that the analysis portion of this paper is not extensive.

We are very pleased that the Referee appreciates the contributions of this study. We truly hope that the new analyses presented here provide the scientific relevance needed for publication in HESS.

Sincerely,

Camila Alvarez Garreton on behalf of co-authors.

[revised manuscript text omitted]

---

## Author Response (AR2)

**"The CAMELS-CL dataset: catchment attributes and meteorology for large sample studies – Chile dataset" by Alvarez-Garreton et al.**

Response letter to anonymous Referees #1, #4 and Margarita Saft.

We thank the Editor, anonymous Referees #1, #4, and Margarita Saft for their constructive comments on our paper.

5 We would like to mention that since our manuscript was submitted, we have developed a dataset explorer (*http://camels.cr2.cl*), which can be used to visualise and download the catchment attributes, time series, and shapefile polygons.

Below we provide responses to each Referee comment. We also provide a revised and marked-up manuscript, where all the changes and additions have been highlighted.

Referee #1:

10 I thank the authors for taking the reviewer comments seriously and substantially modifying this manuscript. The revised version is more interesting and now contains useful analysis methods and discussion for the broader community (as well as specifically for users of CAMELS-CL).

We thank anonymous Referee #1 for the constructive comments and contributions made throughout the revision process.

Referee #3, Margarita Saft:

15 The paper presents a newly available dataset for a region with limited prior data availability. The dataset combines a number of hydrological, climatological, and geographical characteristics allowing for interesting synthesis of these data. I have learned a number of things about Chilean hydrology including regional aspects, even though this did not seem to be the focus of this paper. Another strong side of the paper is that authors clearly describe the limitations of the data and the implications. In my opinion, the paper is relevant for HESS audience, presents a valuable contribution to the field, so I recommend publication 20 subject to addressing comments below.

We thank the Referee for her positive comments and appreciation of the contributions of this study.

Regarding the content and the context of the paper:

The way I see it there are three main aspects in this paper: 1) dataset presentation; 2) characterization of hydrology and related aspects of geography of Chile highlighting regional differences based on novel data; and 3) climate data product comparison 25 in the terrain known to be difficult to get accurate estimates for. The presentation of the paper (abstract/introduction/conclusion) focuses on aspects 1 and 3 and largely omits aspect 2. However, the related parts of the manuscript on their own and in their current state appear somewhat detached. In particular, the core of the dataset is streamflow and catchment data, whereas analysis of uncertainty (in form of spread between different products) is focused solely on the climatic (P and PET) data. Yet aspect 2 (characterizing Chile's hydrology including its social aspects using new data) appears 30 to me to be a logical focus of the paper submitted to HESS, especially since the paper already includes this material. I suggest the authors address this in the abstract/introduction/conclusion.

The Reviewer is right, the abstract and conclusion sections largely focus on presenting the datasets (described in Sect. 3.1) and summarising the findings from the analyses on climatic variables (Sect. 4) and anthropic impacts on catchment response (Sect. 5). On the other hand, we developed a deep discussion on the spatial characteristics of topography, geography, land use, climate 35 and hydrology of Chile (Sect. 3.2) that is not emphasised in a similar degree in the abstract and conclusion sections. Following

the Reviewer suggestion, we revised the abstract, introduction and conclusions to incorporate the findings from the analysis on the spatial distribution of the derived catchment attributes and their inter-relationships.

Specific comments:

3.1.5. Streamflow: Do you include data for all available record or clip it to what is presented at Fig. 3?

5 The dataset includes all available records for each streamflow gauge. Fig. 3 is presented to illustrate the period selected for computing hydrological signatures and climatic indices.

I have tried to download the dataset from the link provided to check that (as I would see including whole records as the strength of the dataset), but the download did not work (zip was 0b size, and individual downloads buttons created a 'File not found or deleted from server' message).

10 The download link was fixed and now the complete dataset can be downloaded from the link provided in the revised manuscript (*http://www.cr2.cl/camels-cl*). In addition, the CAMELS-CL explorer (*http://camels.cr.cl*) can be used to visualise and download the data for individual catchments.

Related question: what are mean and median record lengths (and may be min/max too)?

To address this question, we added 3 new metadata fields to the catchment attributes: 1) gauge_record_start: start date of
15 streamflow records, 2) gauge_record_end: end date of streamflow records, and 3) gauge_n_obs: number of days with valid streamflow records. The statistics are summarized in the Table below:

Table 1: Summary of record length from the 516 streamflow gauges.

|         | days   | months  |
|---------|--------|---------|
| mean    | 10979  | 365.97  |
| median  | 9909   | 330.30  |
| minimum | 192    | 6.60    |
| maximum | 366667 | 1222.23 |

We added a comment on these statistics in revised Sect. 3.1.5.

3.2.5. and Fig.10 b and f: It is surprising to see elasticities < 0 (rainfall increase results in runoff decrease?) and runoff ratios
20 above 1. It would be good to refer to the relevant part of the discussion on >1 runoff ratios in part 4, and comment on the elasticities (is it <0 in Far North? Is it a consequence of opposite trends in P and Q? here).

Following the Reviewer's suggestion, we related the discussion on runoff ratios in Sect. 3.1.5, with the analysis in in Sect. 4, where we argue that runoff ratios above one within near natural catchments are due to underestimation of precipitation.

Thanks for pointing out the results on streamflow elasticity. There was an error in plotting the elasticity in Fig. 10f, which was corrected in the revised manuscript. There are two catchments with negative elasticity values, located in Austral Zone and Southern Patagonia.

The formula used to calculate streamflow elasticity to precipitation corresponds to Eq. 7 in Sankarasubramanian et al., (2001) adapted in Addor et al., (2017) (Table 3):

$$\epsilon = median\left(\frac{Q_t - Q_a}{Q_a} \quad \frac{P_a}{P_t - P_a}\right),$$

where Qt and Pt are annual runoff and precipitation, respectively. Qa and Pa are the long-term (1990-2010) mean annual runoff and precipitation, respectively. This nonparametric approach has a numerical problem when annual precipitation of a single year approaches the long-term mean, causing the elasticity to approach infinity (Sankarasubramanian et al., 2001). In our case, this caused very large negative values for single years, which affected the median of the complete period of analysis. In Fig. 10f, the two negative elasticities (-0.13 and -.03, respectively) are due – in part – to this numerical artefact. Another factor causing these negative elasticities is the use of incomplete streamflow daily records. Based on the discussion on records availability from Fig. 3, we selected catchments with valid daily streamflow records over at least 85% of the period 1990-2010 (explained in Sect. 3.2.5). Since the elasticity is calculated from coincident daily streamflow and precipitation records, its calculation in catchments with missing streamflow records can be problematic. This can be particularly important in snow dominated catchments (delayed runoff response to precipitation) and in catchments with a weak precipitation seasonality (i.e., precipitation falling during the whole year, Fig. 9a close to zero, which is the case for catchments in Austral Zone and Southern Patagonia). A combination of these two issues is causing negative elasticity values in the two southern catchments. This discussion was added to the revised Sect. 3.2.5.

3.1.5. and Table 3 / Elasticity: What method was used to calculate elasticity?

Please refer to our reply above.

3.1.9. and Fig. 4e: Snowmelt data is not shown in the northern and southern catchments, I was wondering whether lack of catchment markers indicates no snowmelt, or no data? Are catchments with no snowmelt were included on the map as orange points, or just not included on the map? I would expect to see some snowmelt in the Patagonia catchments. Can you please comment on the data availability in regard to the current paper study area?

Thanks for pointing this out. The SWE dataset used here covers only the Near North and Central zones. We clarified this in Sect. 3.1.10 and in Fig. 4 caption. All the colored points have data (i.e., orange points represent close to zero values).

Yes, it should be Fig. 9d. The text was corrected in revised Sect. 3.2.4.

3.2.5. Near-natural catchments: Plantations are another factor which can considerably change catchment hydrological signature, it might be beneficial to set some threshold for plantation % as well.

We agree with the Reviewer; forest plantations can significantly affect the hydrologic response of a catchment (we discussed this in Sect. 3.2.3). We set a 20% threshold of forest plantations (the same threshold used for croplands) for selecting near-natural catchments in the revised Sect. 3.2.5, which resulted in 94 selected catchments. The text in Sect. 3.2.5, Figs. 10, 12 and 13 were modified accordingly.

4.1. in particular p.15 lines 28-32: Assessment of the spread is closely related to mean values; may be comment on the absolute errors as well as relative.

A comment was added to revised Sect. 4.1.

p.17 lines 1-8: Nice clear description of the limitations

We thank the Reviewer for her comment.

Histograms in the bottom of the maps (in a number of cases, e.g. Fig.6 a, d, e): Smaller bins near lower tail would be more informative.

If I understood correctly, lower tail refers to lower frequency values, in which case narrower bins would not improve visualization (middle panel in Fig. 6a showed below as an example). We also tried wider bins (right panel in Fig 6a below), but in our opinion, the visualization does not significantly improve. Based on this, the bins were not modified in the revised manuscript.

[Figure]

Fig.6a as an example. Left panel is the figure in the manuscript. Middle and right panels show narrower and wider bins for low frequencies, respectively.

Fig. 7 a, b: It would be good to see bar plot(s), possibly frequency-sorted, and may be even stacked.

We added the bar plots to revised Fig. 7a, b.

Fig. 8: Nice to see no forest cover also presented on the plot – add to the legend?

We added the clarification to the Fig 8 caption.

Fig. 9i: Green column (I assume djf) is missing from the bar chart/legend

The label was added to Fig. 9i.

Table 3: barren_frac unit is missing

The text was corrected in revised Table 3

Referee #4:

The manuscript entitled "The CAMELS-CL dataset: catchment attributes and meteorology for large sample studies – Chile dataset" developed by Alvarez-Garreton C. et al. presents a dataset with daily resolution of several climatic variables. This is of high interest in order to develop climate research, especially in a country as Chile, particularly vulnerable to climate change effects and where water management is a priority for policymakers.

My first thought when I received and accepted to review was, coinciding with the editor, that this manuscript fit perfectly in Earth System Science Data journal, but this is a matter that has to be solved by the editorial team. Nevertheless, the manuscript shows high hydrological interest (mainly for the inclusion of the human intervention on catchments) and has developed a great improvement since its original version following the other referees comments. Despite this, I would like the authors to answer some final concerns.

We thank the Referee for appreciating the contributions of this study and qualifying it of high climatic and hydrological interest.

Major Comments:

Maybe the first concern about this manuscript is that I miss a more detailed explanation about how the authors deal with the
15 fact that the precipitation datasets show projections that display large uncertainty in high and dry areas, as those located in the Far and Near North.

One of the main challenges in studying water resources in Chile is precisely what the Referee mentions: there are no accurate precipitation estimates over the Andes and in arid zones. Permanent efforts are made from national institutions to make best use of the sparse ground data available and improving precipitation estimates. The most updated precipitation national product,
20 developed by the Center for Climate and Resilience Research in a joint project with the National Water Directorate (DGA), is CR2MET (described in Sect. 3.1.6). This product was developed for the National Water Balance project (DGA, 2017) and has been continuously assessed and improved by the evaluation of hydrological predictions from the Variable Infiltration Capacity (VIC) model, and by the inputs from the multidisciplinary group of experts involved in the project.

Having said that, in this work we are not directly addressing the limitations in precipitation estimates. Instead, we assessed the
25 quality of the products that the scientific and technical community most probably will use in hydrological applications within the domain. This was done by: 1) compiling the best available precipitation products for the national territory (described in Sect. 3.1.6) and process them at the catchment scale, and 2) to assess their quality and discuss their limitations by performing an inter-product comparison and analysing the catchments water balance, which has not been done in the domain. With this, we provide estimations of precipitation uncertainty that can be used in hydrological applications (Sect. 4.1).

30 Another point is a matter of scale, but that affects directly the most strong point of this work to this reviewer's eyes: how have been considered Andean peatlands in this work? They play a key role as water reservoirs and their presence modify significantly the water flow, but their dimensions make them "invisible" to many regional to global products (Tachikawa et al. 2011). They're also very sensitive to annual climate variability, which makes me feel that the period used in Zhao et al. (2016) is not enough consider these ecosystems in the model. Finally, they have been traditionally managed by human activity, and
35 their influence in the groundwater flow of high elevated is determinant. Do the authors have considered this?

The Referee rises another important issue we face in northern altiplanic catchments. Andean peatlands largely influence hydrologic response in these catchments. Further, peatlands may be connected by groundwater systems that do not necessarily coincide with surface catchment boundaries. We mentioned this issue in Sect. 4.1, when we discussed the critical role of

groundwater contribution in northern catchments. In the revised Sect. 4.1, we relate this with the critical role of peatlands in catchment response.

We agree with the Reviewer regarding the limitations of the land cover map in accounting for inter-annual variability of altiplanic wetlands. Zhao et al., (2016) performed a detailed seasonal analysis on altiplanic wetlands using 30-m resolution Landsat imagery combined with MODIS images and field photos (see Fig. 13 in Zhao et al., 2016) for 2013 and 2014, however, multiyear analysis was not performed. We mentioned this limitation in revised Sect. 3.2.3.

To account for anthropic intervention within these sensitive ecosystems, we processed the allocated surface and groundwater rights within the altiplanic catchments (Sect. 3.1.10). We highlighted the limitations of these water rights data and the derived attributes in Sect. 3.2.6. A caveat of the human intervention attributes proposed in CAMELS-CL is that the interv_degree (Table 3) only accounts for surface allocated volume within a catchment, while disregarding allocated groundwater volumes. This implies that the selection of near natural catchments does not consider groundwater withdrawals, which can be particularly important in these altiplanic catchments. We added this discussion to revised Sect. 3.2.6.

Minor comments:

- for the whole manuscript, author's guidelines in Physical dimensions and units, point h, says: "The symbol for the decimal marker is the dot. To facilitate reading, numbers may be divided in groups of three using a thin space (e.g. 12 345.6), starting with the ten-thousand digit. Neither dots nor commas are permitted as group separators". Please, correct.

The complete manuscript was revised and corrected.

- p.2 l. 31: I don't really see the point here to talk about the main results of the research and, next, start talking about the study area. Why not directly in the "Study area" section? I would delete these sentences.

The sentences were deleted in the revised Introduction.

- p. 3 l. 15: I miss a reference at the end of the enumeration, concretely for "climate change impacts on the hydrology of the CONUS".

The reference was added.

- p. 4 l. 13: strictly, Kottek et al. (2006) did not applied the Köppen's but the Köppen-Geiger's classification. See Sarricolea P., Herrera-Ossandon M.J., Meseguer-Ruiz O. (2017) Climatic Regionalisation of Continental Chile. Journal of Maps 13(2): 66-73. DOI: 10.1080/17445647.2016.1259592 for further details.

Thanks for the reference. We corrected the text and added the reference to revised Sect.

- p. 8 l. 12: is it possible to know some information about the multivariate regression models? Mainly mean square error and residual.

The computation of the CR2MET products includes several stages. An assessment of the methodology, partially included in DGA (2017), is deeply described in a manuscript still in preparation. The figure that follows may serve to elucidate the referee's query. It shows a contrast of Tmax provided by both CR2MET and ERA-Interim against local records in central-northern Chile (the gridded data of both product are interpolated to the obs sites). In the CR2MET case, the comparison involves a "leave-one-out" cross-validation computation (i.e. the record of the observational site being assessed has not being included

in the product calibration). Two metrics are shown: annual mean bias ("sesgo medio", in left panels) and determination coefficients ("Coef. de determinación", $R^2$), measuring the year-to-year covariance (right).

General speaking, with respect to ERA-Interim, CR2MET significantly reduces the biases and increases interannual correlations, showing the added value of the methodology. Yet, although the performance of CR2MET is quite good in central
5   Chile ($R^2 > 0.7$), it weakens in regions such the Altiplano (north-east). The reduced performance in some regions is due to a poorer agreement between the gridded temperature datasets used as reference (MODIS LST and ERA-Interim) and the observations, leading to multivariate models of lesser predictive capacities.

[Figure]

- p. 10 l. 25: there is a repeated "6", it should be "Figure 6b" nor "Figure 66b".

10  The text was corrected.

- p. 13 l. 10 and l. 13: in English, months start with a capital letter, so it should be written "DJF" and "JJA" when referring to summer and winter.

The text was corrected.

- p. 14 l. 30: I think there is a missing letter: "It should be noted…".

15  The text was corrected.

- p. 19 l. 5: it should be "quartile" instead of "quantile".

The text was corrected.

The text was corrected.

Sincerely,

5    Camila Alvarez Garreton on behalf of co-authors.

[revised manuscript text omitted]

---

## Author Response (AR4)

**"The CAMELS-CL dataset: catchment attributes and meteorology for large sample studies – Chile dataset" by Alvarez-Garreton et al.**

Response letter to Editor Jan Seibert.

Editor comments:
Dear authors,
thanks for your revisions of the HESS-manuscript. I find the manuscript now acceptable for publication in HESS, but the data needs to be available from a website in English.

It would be rather frustrating for HESS readers to see the paper but not being able to access the data from a site in English. After all, the data availability is a key part of this publication. We, thus, should wait with final acceptance until the English version is ready. Thanks for your understanding.

Best regards,
Jan

The CAMELS-CL explorer (http://camels.cr2.cl) is now available in English. We developed this platform with the aim to facilitate a wide range of applications, including research, technical applications, teaching and decision making. The visualisation of the dataset provides a general perspective of the catchments included in CAMELS-CL, including their location and geography, shape, anthropic intervention, topographic and land cover attributes. Further, the user can visualise the hydro-meteorological time series of a selected catchment, with the option of selecting the time step and date range of the plot. From the explorer, all the data for a selected catchment can be downloaded.

For applications requiring the complete dataset, CAMELS-CL data was stored at PANGAEA repository (https://www.pangaea.de) and can be accessed from https://doi.pangaea.de/10.1594/PANGAEA.894885.

Sincerely,

Camila Alvarez Garreton on behalf of co-authors.